# Pearls from Pebbles: Improved Confidence Functions for Auto-labeling

**Harit Vishwakarma**
hvishwakarma@cs.wisc.edu
University of Wisconsin-Madison

**Yi Chen**
yi.chen@wisc.edu
University of Wisconsin-Madison

**Sui Jiet Tay** *
st5494@nyu.edu
NYU Courant Institute

**Satya Sai Srinath Namburi** *
satya.namburi@gehealthcare.com
GE HealthCare

**Frederic Sala**
fredsala@cs.wisc.edu
University of Wisconsin-Madison

**Ramya Korlakai Vinayak**
ramya@ece.wisc.edu
University of Wisconsin-Madison

## Abstract

Auto-labeling is an important family of techniques that produce labeled training sets with minimum manual annotation. A prominent variant, threshold-based auto-labeling (TBAL), works by finding thresholds on a model's confidence scores above which it can accurately automatically label unlabeled data. However, many models are known to produce overconfident scores, leading to poor TBAL performance. While a natural idea is to apply off-the-shelf calibration methods to alleviate the overconfidence issue, we show that such methods fall short. Rather than experimenting with ad-hoc choices of confidence functions, we propose a framework for studying the *optimal* TBAL confidence function. We develop a tractable version of the framework to obtain `Colander` (Confidence functions for Efficient and Reliable Auto-labeling), a new post-hoc method specifically designed to maximize performance in TBAL systems. We perform an extensive empirical evaluation of `Colander` and compare it against methods designed for calibration. `Colander` achieves up to 60% improvement on coverage over the baselines while maintaining error level below $5\%$ and using the same amount of labeled data.

## 1 Introduction

The demand for labeled data in machine learning (ML) is perpetual. Obtaining it is expensive and time-consuming, creating a bottleneck in ML workflows. Threshold-based auto-labeling (TBAL) is a promising solution to obtain high-quality labeled data at low cost [47, 42, 56]. A TBAL system (Fig. 1) takes unlabeled data as input and outputs a labeled dataset. It works iteratively: in each iteration, it acquires human labels for a small chunk of data to train a model, then auto-labels points using the model's predictions where its *confidence scores* are above a certain threshold. The threshold is determined using validation data so that the auto-labeled points meet a desired *accuracy criteria*. The goal is to maximize *coverage*—the fraction of points automatically labeled (out of the total)—while maintaining accuracy. TBAL powers industry products like Amazon SageMaker Ground Truth [47].

The confidence function is critical to the TBAL workflow (Figure 1). Existing TBAL systems rely on common choices like softmax outputs from neural networks [42, 56]. These functions *are not well aligned with the objective of the auto-labeling system*. Using them results in substantially suboptimal coverage (Figure 2(a)). For this reason, we ask:

---

*Work done while at University of Wisconsin-Madison.

38th Conference on Neural Information Processing Systems (NeurIPS 2024).

> What are the right choices of confidence functions for TBAL and how can we obtain them?

An ideal confidence function for auto-labeling will achieve the maximum coverage at a given auto-labeling error tolerance and thus will bring down the labeling cost significantly. Finding such an ideal function, however, is difficult because of the *inherent tension* between accuracy and coverage. The models used in auto-labeling are often highly inaccurate so achieving a certain error guarantee is easier when being conservative in terms of confidence—but this reduces coverage. Conversely, high coverage may appear to require lowering the requirements in confidence—

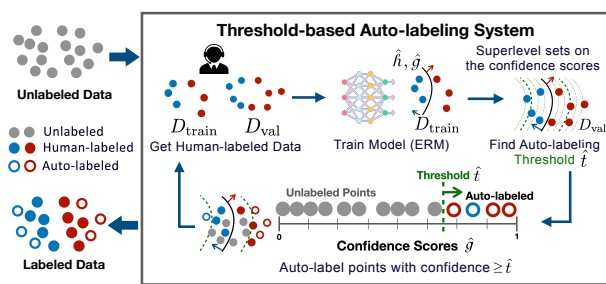

Figure 1: High-level diagram of TBAL system.

dence, but this may easily lead to overshooting the desired error level. This is compounded by the fact that TBAL is iterative, so even small deviations in error levels can cascade in future iterations.

*Overconfidence* may further stymie hopes of balancing accuracy and coverage. While overconfidence is a challenge in general, it is exacerbated in TBAL: since models are trained on a small amount of labeled data, they are often inaccurate, making the problem of designing confidence functions even more challenging. Common choices produce overconfident scores, i.e., high scores for both correct and incorrect predictions [51, 38, 17, 16, 3]. Fig. 2(a) shows that softmax scores are overconfident, resulting in poor auto-labeling performance.

Several methods have been introduced to address overconfidence, including a variety of calibration techniques [12]. Applying these can nevertheless miss out on significant performance (Figure 2(b)) since the calibration goal differs from auto-labeling. From the auto-labeling standpoint, we seek minimum overlap between the correct and incorrect model prediction scores. Other approaches [6, 35] bake the objective of separating scores into model training or use different optimization procedures [64] that encourage separation. We observe that these do not help TBAL either, since, after some point, the model is correct on almost all the training points, making it hard to train it to discriminate between its own correct and incorrect predictions.

We tackle these challenges by ***proposing a framework to learn suitable confidence functions*** for TBAL. In particular, we express the auto-labeling objective as an optimization problem over the space of confidence functions and thresholds. Our framework subsumes existing methods, i.e., they are points in the space of solutions. The resulting method, `Colander` (Confidence functions for Efficient and Reliable Auto-labeling), relies on a practical surrogate to the framework that can be used to learn optimal confidence functions for auto-labeling. Using these learned functions in TBAL can achieve up to 60% improvement in coverage versus baselines like softmax, temperature scaling [12], CRL [35] and FMFP [64]. We summarize our contributions as follows,

1. We propose a principled framework to study the choices of confidence functions suitable for auto-labeling and provide a practical method (`Colander`) to learn confidence functions for efficient and reliable auto-labeling.

2. We systematically study commonly used choices of scoring functions and calibration methods and demonstrate that they lead to poor auto-labeling performance.

3. Through extensive empirical evaluation on real-world datasets, we show that using the confidence scores obtained using our procedure boosts auto-labeling performance significantly in comparison to common choices of confidence functions and calibration methods.

## 2  Background and Motivation

We provide notation, background on TBAL and its relationship to other methods, and describe the importance of confidence functions.

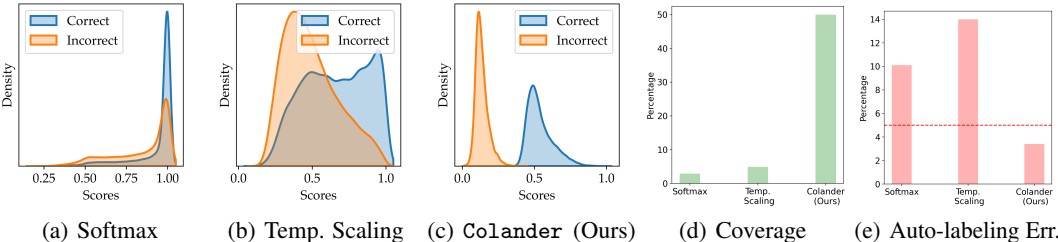

|(a) Softmax|(b) Temp. Scaling|(c) `Colander` (Ours)|(d) Coverage|(e) Auto-labeling Err.|

Figure 2: Scores distributions (Kernel Density Estimates) of a CNN model trained on CIFAR-10 data. (a) softmax scores of the vanilla training procedure (SGD) (b) scores after post-hoc calibration using temperature scaling and (c) scores from our `Colander` procedure applied on the same model. For training the CNN model we use 4000 points drawn randomly and 1000 validation points (of which 500 are used for Temp. Scaling and `Colander`). The test accuracy of the model is 55%. Figures (d) and (e) show the coverage and auto-labeling error of these methods. The dotted-red line corresponds to a user-given error tolerance of 5%.

**Notation.** Let $[m] := \{1, 2, \ldots, m\}$ for any natural number $m$. Let $X_u$ be a set of unlabeled points drawn from some instance space $\mathcal{X}$. Let $\mathcal{Y} = \{1, \ldots, k\}$ be the label space. There is an unknown ground truth labeling function $f^* : \mathcal{X} \to \mathcal{Y}$. Let $\mathcal{O}$ be a *noiseless* oracle that provides the true label for any point $\mathbf{x} \in \mathcal{X}$. Denote the model (hypothesis) class by $\mathcal{H}$, where each $h \in \mathcal{H}$ is a function $h : \mathcal{X} \to \mathcal{Y}$. Each classifier $h$ also has an associated *confidence function* $g : \mathcal{X} \to \Delta^k$ that quantifies the confidence of the prediction by model $h \in \mathcal{H}$ on any data point $\mathbf{x} \in \mathcal{X}$. Here, $\Delta^k$ is a $(k-1)$-dimensional probability simplex. Let $\mathbf{v}[i]$ denote the $i^{\text{th}}$ component for any vector $\mathbf{v} \in \mathbb{R}^d$. For any point $\mathbf{x} \in \mathcal{X}$ the prediction is $\hat{y} := h(\mathbf{x})$ and the associated confidence is $g(\mathbf{x})[\hat{y}]$. The vector $\mathbf{t}$ denotes scores over $k$-classes, and $\mathbf{t}[y]$ denotes its $y^{\text{th}}$ entry, i.e., score for class $y$. Table 3 (in Appendix B.4) contains a summary of the notation.

**Threshold-based auto-labeling.** It seeks to obtain labeled datasets while reducing the labeling burden on humans (Figure 1). The input is a pool of unlabeled data $X_u$. It outputs, for each $\mathbf{x} \in X_u$, a label $\tilde{y} \in \mathcal{Y}$. The output label could be either $y$, from the oracle (human), or $\hat{y}$, from the model. Let $N_u$ be the number of unlabeled points, $A \subseteq [N_u]$ the set of indices of auto-labeled points, and $X_u(A)$ be these points. Let $N_a$ be the size of the auto-labeled set $A$. The *auto-labeling error*, denoted by $\widehat{\mathcal{E}}(X_u(A))$, and the *coverage*, denoted by $\widehat{\mathcal{P}}(X_u(A))$, are defined as follows:

$$\widehat{\mathcal{E}}(X_u(A)) := \frac{1}{N_a} \sum_{i \in A} \mathbb{1}(\tilde{y}_i \neq f^*(\mathbf{x}_i)), \quad \text{and} \quad \widehat{\mathcal{P}}(X_u(A)) := |A|/N_u = N_a/N_u. \quad (1)$$

The goal of an auto-labeling algorithm is to label the dataset so that $\widehat{\mathcal{E}}(X_u(A)) \leq \epsilon_a$ while maximizing coverage $\widehat{\mathcal{P}}(X_u(A))$ for a *user-given error tolerance* parameter $\epsilon_a \in [0, 1]$. As depicted in Figure 1, the TBAL algorithm proceeds iteratively. In each iteration, it queries labels for a subset of unlabeled points from the oracle. It trains a classifier from the model class $\mathcal{H}$ on the oracle-labeled data acquired till that iteration. It then uses the model's confidence scores on the validation data to identify the region in the instance space, where the current classifier is confidently accurate and automatically labels the points in this region. The auto-labeled points are removed from the unlabeled pool. Similarly, to maintain parity between the validation and unlabeled data in the next round, the validation points in the auto-labeling region are removed as well. These steps are executed in a loop until all the data is labeled or the budget to query oracle labels is exhausted.

**Fundamental differences between TBAL, self-training and active learning.** At first glance, TBAL may appear similar to active learning (AL) [46], self-training (ST) [2], and selective classification (SC) [10]. However, as described in [56], it is a fundamentally different technique designed with different goals. Perhaps the most substantial difference is that TBAL's aim is to create accurately labeled datasets, while the goal in AL and ST is to learn the best (in terms of generalization error) possible classifier in a given model class with limited ground truth labels. This difference is most substantial in the settings where AL converges to a bad classifier, e.g., due to incorrect choice of the model class, sampling bias, etc. [56] illustrates this notion with a scenario where TBAL coverage is above 95% while the other techniques average around 20%. See Appendix A.1 for details.

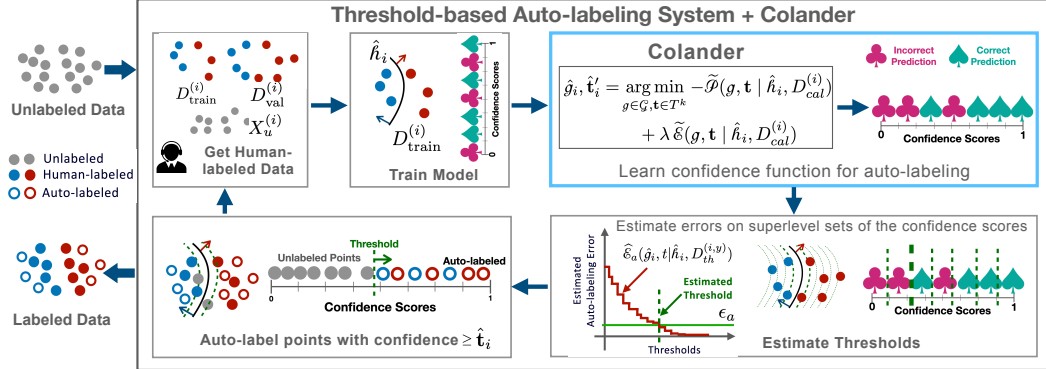

Figure 3: Threshold-based Auto-labeling with `Colander`: takes unlabeled data as input, selects a small subset of data points, and obtains human labels for them to create $D_{\text{train}}^{(i)}$ and $D_{\text{val}}^{(i)}$ for the $i$th iteration. Trains model $\hat{h}_i$ on $D_{\text{train}}^{(i)}$. In contrast to the standard TBAL procedure, here we randomly split $D_{\text{val}}^{(i)}$ into two parts, $D_{\text{cal}}^{(i)}$ and $D_{\text{th}}^{(i)}$. `Colander` kicks in, takes $\hat{h}_i$ and $D_{\text{cal}}^{(i)}$ as input and learns a coverage maximizing confidence function $\hat{g}_i$ for $\hat{h}_i$. Using $D_{\text{th}}^{(i)}$ and $\hat{g}_i$ auto-labeling thresholds $\hat{\mathbf{t}}_i$ are determined to ensure the auto-labeled data has error at most $\epsilon_a$ (a user-given parameter). After obtaining the thresholds the rest of the steps are the same as standard TBAL. The whole workflow runs until all the data is labeled or another stopping criterion is met.

**Problems with confidence functions in TBAL.**     The success of TBAL hinges on the ability of the classifier's confidence scores to distinguish between correct and incorrect labels. Prior works on TBAL [56, 42] train the model with stochastic gradient descent (SGD) and use the softmax output of the model as confidence scores, which are known to be overconfident [38]. A natural choice to mitigate this problem is to use post-hoc calibration techniques, e.g., temperature scaling [12]. We evaluate these choices by running TBAL for a single round on the CIFAR-10 [24] dataset with a SimpleCNN model with 5.8M parameters [20] with error threshold $5\%$. Details are in Appendix A.2.

In Figures 2(d) and 2(e) we observe that using softmax scores from the classifier only produces $2.9\%$ coverage while the error threshold is violated with $10\%$ error. Using temperature scaling only increases the coverage marginally to $4.9\%$ and still violates the threshold with error $14\%$. Looking closer at the scores for correct versus incorrect examples on validation data, we observe a large overlap for softmax (Figure 2(a)) and a marginal shift with considerable overlap for temperature scaling (Figure 2(b)). To overcome this challenge, we propose a novel framework (Section 3) to learn such confidence functions in a principled way. Our method in this example can achieve $50\%$ coverage with an error of $3.4\%$ within the desired threshold (Figure 2(c)).

## 3   Proposed Method (`Colander`)

The observations in Figure 2(a) and 2(b) suggest that fixed choices of confidence functions can leave significant coverage on the table. To find a better choice in a principled manner, we develop a framework based on auto-labeling objectives—maximizing coverage while having bounded auto-labeling error. We instantiate it by using empirical estimates and easy-to-optimize surrogates. We use the overall TBAL workflow from [56] and introduce our method to replace the confidence (scoring) function after training the classifier.

### 3.1   Auto-labeling optimization framework

In any iteration of TBAL, we have a model $h$ trained on a subset of data labeled by the oracle. This model may not be highly accurate. However, it could be accurate in some regions of the instance space, and with the help of a confidence function $g$, we want to identify the points where the model is correct and auto-label them. As we saw earlier, arbitrary choices of $g$ perform poorly on this task. Instead, we propose a framework to find the right function from a sufficiently rich family.

**Optimal confidence function.** To find a confidence function aligned with our objective, we consider a space of thresholds $T$ (e.g. $[0, 1]$) and functions $\mathcal{G} : \mathcal{X} \to T^k$, where $T^k$ is the $k-$dimensional product space of $T$. We express the auto-labeling objective as an optimization problem (P1):

$$\underset{g \in \mathcal{G}, \mathbf{t} \in T^k}{\arg\max} \quad \mathcal{P}(g, \mathbf{t} \mid \hbar) \quad \text{s.t.} \quad \mathcal{E}(g, \mathbf{t} \mid \hbar) \leq \epsilon_a. \tag{P1}$$

Here $\mathcal{P}(g, \mathbf{t}|\hbar)$ and $\mathcal{E}(g, \mathbf{t} \mid \hbar)$ are the population level coverage and auto-labeling error which are defined as follows,

$$\mathcal{P}(g, \mathbf{t} \mid \hbar) := \mathbb{P}_{\mathbf{x}}\big(g(\mathbf{x})[\hat{y}] \geq \mathbf{t}[\hat{y}]\big) \quad \text{and} \quad \mathcal{E}(g, \mathbf{t} \mid \hbar) := \mathbb{P}_{\mathbf{x}}\big(y \neq \hat{y} \mid g(\mathbf{x})[\hat{y}] \geq \mathbf{t}[\hat{y}]\big). \tag{2}$$

The optimal $g^\star$ and $\mathbf{t}^\star$ that achieve the maximum coverage while satisfying the auto-labeling error constraint belong to the solution(s) of this optimization problem.

## 3.2 Practical method to learn confidence functions

The framework provides a theoretical characterization of the optimal confidence functions and thresholds for TBAL. However, it is impractical since the distributions and $f^\star$ are unknown. Next, we give a practical method based on the above framework to learn confidence functions for TBAL.

**Empirical optimization problem.** Since we do not know the distributions of $\mathbf{x}$ and $f^\star$, we use estimates of coverage and auto-labeling errors on a fraction of validation data to solve the optimization problem. Let $D$ be some finite number of labeled samples, and then the empirical coverage and auto-labeling error are defined as follows,

$$\widehat{\mathcal{P}}(g, \mathbf{t} \mid \hbar, D) := \frac{1}{|D|} \sum_{(\mathbf{x}, y) \in D} \mathbb{1}\big(g(\mathbf{x})[\hat{y}] \geq \mathbf{t}[\hat{y}]\big), \tag{3}$$

$$\widehat{\mathcal{E}}(g, \mathbf{t} \mid \hbar, D) := \frac{\sum_{(\mathbf{x}, y) \in D} \mathbb{1}\big(y \neq \hat{y} \wedge g(\mathbf{x})[\hat{y}] \geq \mathbf{t}[\hat{y}]\big)}{\sum_{(\mathbf{x}, y) \in D} \mathbb{1}\big(g(\mathbf{x})[\hat{y}] \geq \mathbf{t}[\hat{y}]\big)}. \tag{4}$$

We randomly split the validation data into two parts $D_{\text{cal}}$ and $D_{\text{th}}$ and use $D_{\text{cal}}$ to compute $\widehat{\mathcal{P}}(g, \mathbf{t} \mid \hbar, D_{\text{cal}})$ and $\widehat{\mathcal{E}}(g, \mathbf{t} \mid \hbar, D_{\text{cal}})$. Using these estimates, we now seek to solve the following problem,

$$\underset{g \in \mathcal{G}, \mathbf{t} \in T^k}{\arg\max} \quad \widehat{\mathcal{P}}(g, \mathbf{t} \mid \hbar, D_{\text{cal}}) \quad \text{s.t.} \quad \widehat{\mathcal{E}}(g, \mathbf{t} \mid \hbar, D_{\text{cal}}) \leq \epsilon_a. \tag{P2}$$

Nevertheless, the presence of 0-1 variables means the problem remains challenging.

**Surrogate optimization problem.** To make the optimization (P2) tractable using gradient-based methods, we introduce differentiable surrogates for the 0-1 variables. Let $\sigma(\alpha, z) := 1/(1 + \exp(-\alpha z))$ denote the sigmoid function on $\mathbb{R}$ with scale parameter $\alpha \in \mathbb{R}$. It is easy to see that, for any $g, y$ and $\mathbf{t}$, $g(\mathbf{x})[y] \geq \mathbf{t}[y] \iff \sigma(\alpha, g(\mathbf{x})[y] - \mathbf{t}[y]) \geq 1/2$. Using this fact, we define the following surrogates of the auto-labeling error and coverage:

$$\widetilde{\mathcal{P}}(g, \mathbf{t}|\hbar, D_{\text{cal}}) := \frac{1}{|D_{\text{cal}}|} \sum_{(\mathbf{x}, y) \in D_{\text{cal}}} \sigma\big(\alpha, g(\mathbf{x})[\hat{y}] - \mathbf{t}[\hat{y}]\big), \tag{5}$$

$$\widetilde{\mathcal{E}}(g, \mathbf{t} \mid \hbar, D_{\text{cal}}) := \frac{\sum_{(\mathbf{x}, y) \in D_{\text{cal}}} \mathbb{1}\big(y \neq \hat{y}\big) \, \sigma\big(\alpha, g(\mathbf{x})[\hat{y}] - \mathbf{t}[\hat{y}]\big)}{\sum_{(\mathbf{x}, y) \in D_{\text{cal}}} \sigma\big(\alpha, g(\mathbf{x})[\hat{y}] - \mathbf{t}[\hat{y}]\big)}, \tag{6}$$

and the surrogate optimization problem as follows,

$$\underset{g \in \mathcal{G}, \mathbf{t} \in T^k}{\arg\min} \quad -\widetilde{\mathcal{P}}(g, \mathbf{t} \mid \hbar, D_{\text{cal}}) + \lambda \widetilde{\mathcal{E}}(g, \mathbf{t} \mid \hbar, D_{\text{cal}}) \tag{P3}$$

Here, $\lambda \in \mathbb{R}^+$ is the penalty term controlling the relative importance of the auto-labeling error and coverage. We tune it with the procedure discussed in Section 4.3. The gap between the surrogate and actual coverage diminishes as $\alpha \to \infty$. We discuss this in the Appendix.

**Choice of $\mathcal{G}$.** Our framework is flexible with respect to the choice of function class $\mathcal{G}$. In this work, we use neural networks with at least two layers on model class $\mathcal{H}$. We use representations from the last two layers as input for the functions in $\mathcal{G}$ (Figure 4). Let $\mathbf{z}^{(1)}(\mathbf{x}; \hbar) \in \mathbb{R}^k$ and $\mathbf{z}^{(2)}(\mathbf{x}; h) \in \mathbb{R}^{d_2}$ be the outputs of the last and the second-last layer of the net $h$ for input $\mathbf{x}$ and let $\mathbf{z}(\mathbf{x}; h) := [\mathbf{z}^{(1)}(\mathbf{x}; \hbar), \mathbf{z}^{(2)}(\mathbf{x}; \hbar)]$ denote the concatenation. This input is passed to network $\mathcal{G}_{nn_2} : \mathbb{R}^{k+d_2} \mapsto \Delta^k$; it outputs confidence scores for the $k$ classes. Specifically $g$ is defined as $g(\mathbf{x}) := \mathrm{softmax}\big(\mathbf{W}_2 \mathrm{tanh}(\mathbf{W}_1 \mathbf{z}(\mathbf{x}; \hbar))\big)$. Here $\mathbf{W}_1 \in \mathbb{R}^{(k+d_2) \times 2(k+d_2)}$ and $\mathbb{R}^{2(k+d_2) \times k}$ are the learnable weight matrices. As usual, for $\mathbf{v} \in \mathbb{R}^d$, $\mathrm{softmax}(\mathbf{v})[i] := \exp(\mathbf{v}[i]) / (\sum_j \exp(\mathbf{v}[j]))$ and $\mathrm{tanh}(\mathbf{v})[i] := (\exp(2\mathbf{v}[i]) - 1) / (\exp(2\mathbf{v}[i]) + 1)$.

We emphasize, `Colander` can use any function class for $\mathcal{G} : \mathcal{X} \to T^k$. Here, we chose 2-layer nets and successfully used the same across all experiments, thus we may not need an exhaustive architecture search. Intuitively, we do not need a large network for $g$ since $\hbar$ already performs the heavier representation learning work. As a result, simple models are preferable to avoid overfitting and to reduce training time since post-hoc methods should be fast.

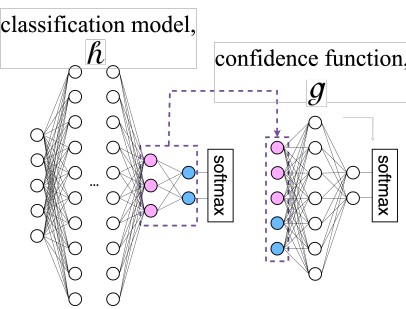

Figure 4: Our choice of $g$ function.

**Solving the surrogate optimization.** The optimization problem (P3) is nonconvex. Nevertheless, it is differentiable and we can apply gradient-based methods. We solve for $g$ and $\mathbf{t}$ simultaneously using Adam [23]. Training details, including hyperparameters, are deferred to the Appendix.

### 3.3 TBAL procedure with `Colander`

We take the workflow of TBAL and plugin our method `Colander` to learn the new confidence function and threshold. We discuss the updated workflow below and place the detailed Algorithms 1 and 2 in the Appendix B due to space constraints.

**1. Initialization.** First, select $n_s$ points randomly from $X_u$ and obtain human labels for them to create initial training data $D_{\mathrm{train}}^{(1)}$. This is written as RANDOMQUERY$(X_u, n_s)$ in Algorithm 1. The procedure RANDOMQUERY$(X_u, n_s)$ selects $n_s$ points randomly from $X_u$ and obtains human labels for them to create $D_{\mathrm{train}}^{(1)}$. The steps of this procedure are detailed in Algorithm 4 in the Appendix B.

**2. Train classification model.** After obtaining human-labeled training data $D_{\mathrm{train}}^{(i)}$ for the current round $i$, the procedure TRAINMODEL$(\mathcal{H}, D_{\mathrm{train}}^{(i)})$ trains a model from model class $\mathcal{H}$ on the training data $D_{\mathrm{train}}^{(i)}$. Any training procedure can be used here. We use methods listed in Section **??** for model training. This step outputs a model $\hat{h}_i$ trained on $D_{\mathrm{train}}^{(i)}$. Note, this model $\hat{h}_i$ may not be highly accurate due to several factors such as the small amount of training data, the choice of $\mathcal{H}$, the training algorithm, and its hyperparameters. Indeed, TBAL does not expect the model to have high accuracy but it aims to identify and auto-label points where the model's accuracy is at least $1 - \epsilon_a$.

**3. Learn new confidence function using `Colander`.** The model $\hat{h}_i$ obtained in the previous step also produces softmax scores that can be used for auto-labeling. However, as we saw earlier in Section 2, using these scores may lead to poor auto-labeling performance. Thus, we plug in our procedure `Colander` to learn new scores designed to auto-label as many points as possible with the current model $\hat{h}_i$ while respecting the error constraint. We first randomly splits the validation data $D_{\mathrm{val}}^{(i)}$ into $D_{\mathrm{cal}}^{(i)}$ and $D_{\mathrm{th}}^{(i)}$ using procedure RANDOMSPLIT$(D_{\mathrm{val}}^{(i)}, \nu)$. The part $D_{\mathrm{cal}}^{(i)}$ has a fraction $\nu$ of the points from $D_{\mathrm{val}}^{(i)}$. Then we consider problem P3 with $\hat{h}_i$ and $D_{\mathrm{cal}}^{(i)}$. We solve it to obtain the post-hoc confidence function $\hat{g}_i$, which is expected to provide the most appropriate scores for TBAL.

We get thresholds $\hat{t}'_i$ as output from `Colander`, and it is tempting to use these along with $\hat{g}_i$ for auto-labeling. However, these thresholds may violate the auto-labeling error constraint as they

are obtained by solving the relaxed optimization problem. Thus, it is crucial to estimate reliable thresholds $\hat{\mathbf{t}}_i$ from the held-out data $D_{\text{th}}^{(i)}$ to ensure the auto-labeling error constraint is not violated.

**4. Threshold estimation.** The scores from the new confidence function $\hat{g}_i$ on $D_{\text{th}}^{(i)}$ are used to estimate auto-labeling thresholds in Algorithm 2. This procedure finds thresholds for each class separately. It first splits the points in $D_{\text{th}}^{(i)}$ according to the ground truth class into subsets $D_{\text{th}}^{(i,y)}$. Then, for each class $y$, it finds the auto-labeling threshold $\hat{\mathbf{t}}[y]$ by selecting the minimum threshold $t$ such that the estimate of auto-labeling error for class $y$, $\widehat{\mathcal{E}}_y(\hat{g}_i, t|\hat{h}_i, D_{\text{th}}^{(i,y)})$ plus a confidence interval $\widehat{\zeta}(\widehat{\mathcal{E}}_y(\hat{g}_i, t|\hat{h}_i, D_{\text{th}}^{(i,y)}))$ estimated on points in $D_{\text{th}}^{(i,y)}$ having scores above $t$, is at most the given error tolerance $\epsilon_a$. Here $\widehat{\zeta}(z) = C_1\sqrt{z(1-z)}$ for $z \in [0,1]$ and $C_1 \geq 0$ is a hyperparameter.

**5. Auto-labeling.** This is a simple step. We compute the scores on the remaining unlabeled data $X_u^{(i)}$ using the function $\hat{g}_i$ and any point $\mathbf{x} \in X_u^{(i)}$ having score above $\hat{\mathbf{t}}[\hat{y}]$ is assigned auto-label $\hat{y} = \hat{h}_i(\mathbf{x})$, and the points that did not meet this criterion remain unlabeled.

**6. Remove auto-labeled points.** The points that got auto-labeled in the previous steps are removed from the unlabeled pool. To make the validation data consistent with this unlabeled pool for the next round, the points in the validation data that fall into the auto-labeling region are also removed.

**7. Get more human-labeled data.** Lastly, it calls the procedure ACTIVEQUERY$(\hat{h}_i, X_u^{(i)}, n_b)$ to select $n_b$ points from the remaining unlabeled pool using an active learning strategy. This newly acquired human-labeled data is added to the training data $D_{\text{train}}^{(i)}$. The details of the querying strategy are in Algorithm 3 Appendix B. Note, that the TBAL procedure is flexible to work with any choice of active querying strategy. We pick a simple strategy based on random sampling from the points where the classifier is most uncertain. To avoid confounding in TBAL with other scores we use the softmax scores from the classifier to determine uncertainty here.

The procedure then moves to step 2 and runs the loop until there are no more unlabeled points left or it has queried the stipulated number of human labels $N_t$.

## 4  Empirical Evaluation

We validate the following claims through extensive empirical evaluation,

**C1.** `Colander` learns better confidence functions for auto-labeling compared to standard training and common post-hoc methods that seek to mitigate the overconfidence problem. Using it in TBAL can boost the coverage significantly while keeping the auto-labeling error low.

**C2.** `Colander` is independent of any particular train-time method and thus should help improve the performance when coupled with different train-time methods.

### 4.1  Baselines

We examine several train-time and post-hoc methods that improve confidence functions from the calibration and ordinal ranking perspectives. Details of these methods are in the Appendix C.7.

**Train-time methods.** We use the following methods for training the model $\hat{h}$. *Vanilla* neural networks are trained with the cross-entropy loss using stochastic gradient descent (SGD) [1, 4, 12]. *Squentropy* [19] adds the average square loss over the incorrect classes to the cross-entropy loss to improve the calibration and accuracy of the model. *Correctness Ranking Loss (CRL)* [35] aligns the confidence scores of the model with the ordinal rankings criterion via regularization. *FMFP* [64] aligns confidence scores with the ordinal rankings criterion by using sharpness-aware minimization (SAM) [7] in lieu of SGD.

**Post-hoc methods.** We use the following methods for learning (or updating) the confidence function $\hat{g}$ after learning $\hat{h}$. *Temperature scaling* [12] is a variant of Platt scaling [41]. It rescales the logits by a learnable scalar parameter. *Top-Label Histogram-Binning* [14] builds on the histogram-binning method [62] and focuses on calibrating the scores of the predicted label assigned to unlabeled points. *Scaling-Binning* [28] applies temperature scaling and then bins the confidence function values. *Dirichlet Calibration* [26] models the distribution of predicted probability vectors separately on

| Dataset | Model $\hbar$ | $N$ | $N_u$ | $K$ | $N_t$ | $N_v$ | $N_{\text{hyp}}$ | Modality | Preprocess | Dimension |
|---|---|---|---|---|---|---|---|---|---|---|
| MNIST | LeNet-5 | 70k | 60k | 10 | 500 | 500 | 500 | Image | None | $1 \times 28 \times 28$ |
| CIFAR-10 | CNN | 50k | 40k | 10 | 10k | 8k | 2k | Image | None | $3 \times 32 \times 32$ |
| Tiny-Imagenet | MLP | 110k | 90k | 200 | 10k | 8k | 2k | Image | CLIP | 512 |
| 20 Newsgroup | MLP | 11.3k | 9k | 20 | 2k | 1.6k | 600 | Text | FlagEmb. | 1,024 |

Table 1: Details of the dataset and model we used to evaluate the performance of our method and other calibration methods. For the Tiny-Imagenet and 20 Newsgroup datasets, we use CLIP and FlagEmbedding, respectively, to obtain the embeddings of these datasets and conduct auto-labeling on the embedding space. For Tiny-Imagenet, we use a 3-layer perceptron with 1,000, 500, 300 neurons on each layer as model $\hbar$; for 20 Newsgroup, we use a 3-layer perceptron with 1,000, 500, 30 neurons on each layer as model $\hbar$.

instances of each class and assumes Dirichlet class conditional distributions. *Adaptive Temperature Scaling* [21] builds on top of temperature scaling and considers that different samples contribute to the calibration error differently. Each train-time method is piped with a post-hoc method, yielding a total of $4 \times 6 = 24$ methods.

## 4.2 Datasets and models

We evaluate the performance of auto-labeling on four datasets. Each is paired with a model for auto-labeling: *MNIST* [30] is a hand-written digits dataset. We use the LeNet [31] for auto-labeling. *CIFAR-10* [24] is an image dataset with 10 classes. We use a CNN with approximately 5.8M parameters [20] for auto-labeling. *Tiny-ImageNet* [29] is an image dataset comprising 100K images across 200 classes. We use CLIP [43] to derive embeddings for the images in the dataset and use an MLP model. *20 Newsgroups* [34] is a natural language dataset comprising around 18K news posts across 20 topics. We use the FlagEmbedding [58] to obtain text embeddings and use an MLP model.

## 4.3 Hyperparameter search and evaluation

The complexity of TBAL workflow and lack of labeled data make hyperparameter search and evaluation challenging. Similar challenges have been observed in active learning [32]. We discuss our practical approach and defer the details to Appendix C.10 and code[2].

**Hyperparameter search.** We run only the first round of TBAL with each method using a hyper-parameter combination 5 times and measure the mean auto-labeling error and mean coverage on $D_{\text{hyp}}$, which represents a small part of the held-out human-labeled data. We pick the combination that yields the lowest average auto-labeling error while maximizing the coverage. We first find the best hyperparameters for each train-time method, fix those, and then search the hyperparameters for the post-hoc methods. Note that the best hyperparameter for a post-hoc method depends on the training-time method that it pipes to. The hyperparameter search spaces are in the Appendix C; and the selected values used for each setting are in the supplementary material.

**Performance evaluation.** After fixing the hyper-parameters, we run TBAL with each combination of train-time and post-hoc method on full $X_u$ of size $N$, with a fixed budget of $N_t$ labeled training samples and $N_v$ validation samples. The details of these values for each dataset are in Table 1 in Appendix C. Here, we know the ground truth labels for the points in $X_u$, so we measure the auto-labeling error and coverage as defined in (1) and report them in Table 2.

## 4.4 Results and discussion

Our findings, shown in Table 2, are:

*C1: `Colander` **improves TBAL performance.*** Our approach aims to optimize the confidence function to maximize coverage while minimizing errors. When applied to TBAL, we expect it to yield substantial coverage enhancement and error reduction compared to vanilla training and softmax scores. Indeed, the results in Table 2 corresponding to the vanilla training match our expectations. We see *across all data settings*, our method achieves *significantly higher coverage* while keeping auto-labeling error below the tolerance level of 5%. The improvements are even more pronounced when

---

[2]https://github.com/harit7/TBAL-Colander-NeurIPS-24

| Train-time | Post-hoc | MNIST | | CIFAR-10 | | 20 Newsgroups | | Tiny-ImageNet | |
|---|---|---|---|---|---|---|---|---|---|
| | | Err ($\downarrow$) | Cov ($\uparrow$) | Err ($\downarrow$) | Cov ($\uparrow$) | Err ($\downarrow$) | Cov ($\uparrow$) | Err ($\downarrow$) | Cov ($\uparrow$) |
| Vanilla | Softmax | **4.1**±0.7 | 85.0±2.5 | 4.8±0.2 | 14.0±2.1 | 6.0±0.6 | 48.2±1.6 | 11.1±0.3 | 32.6±0.5 |
| | TS | 7.8±0.6 | 94.2±0.5 | 7.3±0.3 | 23.2±0.7 | 9.7±0.6 | 60.7±2.3 | 16.3±0.5 | 37.4±1.5 |
| | Dirichlet | 7.9±0.7 | 93.2±2.2 | 7.7±0.5 | 22.4±1.2 | 9.4±0.9 | 59.4±1.8 | 17.1±0.4 | 33.3±2.0 |
| | SB | 6.7±0.5 | 92.6±1.5 | 6.1±0.4 | 18.6±1.1 | 8.1±0.6 | 58.1±1.8 | 15.7±0.6 | 35.4±1.2 |
| | Top-HB | 7.4±1.4 | 93.1±3.6 | 6.0±0.7 | 15.6±1.9 | 9.2±1.0 | 59.0±2.0 | 16.6±0.5 | 37.6±2.2 |
| | AdaTS | 7.5±0.9 | 92.8±2.0 | 8.6±0.6 | 16.9±1.0 | 9.6±1.1 | 61.8±3.3 | 15.9±0.7 | 36.7±1.9 |
| | **Ours** | 4.2±1.5 | **95.6**±1.4 | **3.0**±0.2 | **78.5**±0.2 | **2.5**±1.1 | **80.6**±0.7 | **1.4**±2.1 | **59.2**±0.8 |
| CRL | Softmax | 4.7±0.4 | 86.0±4.5 | 5.2±0.3 | 15.9±0.8 | 5.8±0.5 | 48.3±0.3 | 10.4±0.4 | 32.5±0.6 |
| | TS | 8.0±0.8 | 94.8±0.8 | 6.8±0.8 | 20.3±1.1 | 9.5±1.0 | 61.7±1.6 | 15.8±0.6 | 37.4±1.7 |
| | Dirichlet | 8.6±0.6 | 93.1±1.6 | 7.7±0.2 | 20.9±1.1 | 8.7±0.9 | 58.0±1.4 | 16.3±0.4 | 33.1±1.9 |
| | SB | 7.4±0.8 | 93.1±2.7 | 5.9±0.9 | 17.9±1.5 | 8.9±1.1 | 57.9±3.9 | 15.0±0.4 | 35.5±1.2 |
| | Top-HB | 7.7±0.8 | 94.1±1.5 | 4.4±0.5 | 12.3±0.4 | 8.8±1.0 | 58.8±2.7 | 16.5±0.5 | 38.9±1.6 |
| | AdaTS | 7.8±0.7 | 94.3±1.2 | 8.8±0.4 | 17.1±1.2 | 9.1±0.8 | 60.8±1.9 | 16.2±0.4 | 38.9±1.2 |
| | **Ours** | **4.5**±1.4 | **95.6**±1.3 | **2.2**±0.6 | **77.9**±0.2 | **1.8**±1.2 | **81.3**±0.5 | **2.8**±2.1 | **61.2**±1.4 |
| FMFP | Softmax | 4.8±0.8 | 84.2±4.1 | 4.9±0.4 | 15.6±1.7 | 5.4±0.7 | 45.4±1.9 | 10.5±0.3 | 32.4±1.4 |
| | TS | 8.0±0.6 | 95.3±1.6 | 6.5±0.3 | 21.0±1.5 | 9.5±0.5 | 57.7±2.2 | 16.2±1.1 | 37.7±1.8 |
| | Dirichlet | 8.2±1.3 | 94.0±2.2 | 6.9±0.4 | 21.7±1.2 | 8.9±1.0 | 56.6±2.4 | 17.4±0.8 | 33.0±1.8 |
| | SB | 7.2±1.1 | 93.1±2.3 | 6.1±0.5 | 19.5±1.0 | 8.6±0.4 | 55.8±1.3 | 15.5±0.6 | 36.1±0.5 |
| | Top-HB | 7.1±0.6 | 93.3±4.9 | 5.2±0.5 | 14.2±2.4 | 9.0±0.7 | 57.9±2.4 | 16.2±0.4 | 37.4±1.1 |
| | AdaTS | 7.6±0.4 | 94.1±1.0 | 7.2±0.7 | 27.5±1.5 | 8.7±0.9 | 56.7±2.7 | 16.3±0.6 | 37.6±1.7 |
| | **Ours** | **4.6**±0.8 | **95.7**±0.2 | **3.0**±0.4 | **77.4**±0.2 | **2.5**±0.9 | **80.8**±0.6 | **1.8**±2.0 | **60.8**±1.4 |
| Squentropy | Softmax | **3.7**±1.0 | 88.2±3.9 | 5.2±0.5 | 21.2±1.8 | 4.6±0.4 | 52.0±1.2 | 7.8±0.3 | 36.2±0.8 |
| | TS | 6.2±1.1 | 95.6±0.9 | 6.9±0.6 | 28.2±2.5 | 8.3±0.6 | 66.6±1.4 | 13.3±0.1 | 44.9±1.0 |
| | Dirichlet | 6.5±1.2 | 95.9±0.8 | 7.3±0.3 | 29.4±1.1 | 7.8±0.6 | 64.0±1.3 | 14.1±0.3 | 42.5±0.7 |
| | SB | 6.0±0.8 | 95.3±1.2 | 6.2±0.4 | 23.8±1.9 | 7.8±0.7 | 63.0±2.9 | 13.0±0.5 | 45.2±2.0 |
| | Top-HB | 5.3±0.4 | 96.4±0.9 | 4.3±0.5 | 15.8±1.4 | 8.2±0.8 | 66.5±2.2 | 13.7±0.1 | 45.9±1.4 |
| | AdaTS | 6.6±1.0 | 95.7±1.0 | 7.6±0.3 | 22.6±1.2 | 7.4±0.6 | 64.7±2.6 | 14.0±0.3 | 46.1±0.7 |
| | **Ours** | 4.1±0.8 | **97.2**±0.5 | **2.3**±0.5 | **79.0**±0.3 | 3.3±0.8 | **82.9**±0.4 | **0.6**±0.2 | **66.5**±0.7 |

Table 2: In every round the error was enforced to be below 5%; 'TS' stands for Temperature Scaling, 'SB' stands for Scaling Binning, 'Top-HB' stands for Top-Label Histogram Binning. 'AdaTS' stands for Adaptive Temperature Scaling. The column Err stands for auto-labeling error and Cov stands for coverage. Each cell value is mean ± std. deviation on 5 repeated runs with different random seeds.

the datasets are more complex than MNIST. Also consistent with our expectation and observations in Figure 2(b), the post-hoc calibration methods improve the coverage over using softmax scores but at the cost of slightly higher error. While they are reasonable choices to apply in the TBAL pipeline, they fall short of maximally improving TBAL performance due to the misalignment of goals.

***C2: `Colander` is compatible with and improves over other train-time methods.*** Our method is compatible with various choices of train-time methods, and if a train-time method (Squentropy here) provides a *better* model relative to another train-time method (e.g., Vanilla), then our method exploits this gain and pushes the performance even further. Across different train-time methods, we do not see significant differences in the performance, except for Squentropy. Using Squentropy with softmax improves the coverage by as high as 6-7% while dropping the auto-labeling error in contrast to using softmax scores obtained with other train-time methods for the Tiny-ImageNet setting. This is unexpected: Squentropy adds the average square loss over the incorrect classes as a regularizer, and it has offered better accuracy and calibration compared to training with cross-entropy loss.

***Train-time methods designed for ordinal ranking objective perform poorly in auto-labeling.*** CRL and FMFP are state-of-the-art methods designed to produce scores aligned with the ordinal ranking criteria. Ideally, if the scores satisfy this criterion, TBAL's performance would improve. However, we do not see any significant difference from the Vanilla method. Similar to the other baselines, their

evaluation is focused on models trained on large amounts of data. But, in TBAL, we have less data for training. The training error goes to zero after some rounds, and no information is left for the CRL loss to distinguish between correct and incorrect predictions (i.e., count SGD mistakes). On the other hand, FMFP is based on a hypothesis that training models using Sharpness Aware Minimizer (SAM) could lead to scores satisfying the ordinal ranking criteria. However, this phenomenon is still not well understood, especially in settings like ours with limited training data.

## 5 Related Work

**Data labeling.** We briefly discuss prominent methods for labeling. Crowdsourcing [45, 50] uses a crowd of non-experts to complete a set of labeling tasks. Works in this domain focus on denoising the obtained information, modeling label errors, and designing effective labeling tasks [11, 22, 33, 53, 52, 54, 5]. Weak supervision (WS), in contrast, emphasizes labeling through multiple inexpensive but noisy sources, not necessarily human [44, 48, 55, 18, 49, 60, 63]. Works such as [44, 8] concentrate on binary or multi-class labeling, while [48, 55] extend WS to structured prediction tasks.

Auto-labeling occupies an intermediate position between weak supervision and crowdsourcing in terms of human dependency. It aims to minimize costs to obtain human labels while generating high-quality labeled data using a specific model. [42] use a TBAL-like algorithm and explore the cost of training for auto-labeling with large-scale model classes. Recent work [56] theoretically analyzes the sample complexity of validation data required to guarantee the quality of auto-labeled data.

**Overconfidence and calibration.** The issue of overconfidence [51, 38, 16, 3] is detrimental in several applications (such as robustness to out-of-distribution points [59, 57]), including ours. Many solutions have emerged to mitigate the overconfidence and miscalibration problems. Gawlikowski et al. [9] provide a comprehensive survey on uncertainty quantification and calibration techniques for neural networks. Guo et al. [12] evaluated a variety of solutions ranging from the choice of network architecture, model capacity, weight decay regularization [25], histogram-binning and isotonic regression [61, 62] and temperature scaling [41, 39] which they found to be the most promising solution. The solutions fall into two broad categories: train-time and post-hoc. Train-time solutions modify the loss function, include additional regularization terms, or use different training procedures [27, 37, 36, 19]. On the other hand, post-hoc methods such as top-label histogram-binning [13], scaling binning [28], Dirichlet calibration [26] calibrate the scores directly or learn a model that corrects miscalibrated confidence scores.

**Beyond calibration.** While calibration aims to match the confidence scores with a probability of correctness, it is not the precise solution to the overconfidence problem in many applications, including our setting. The desirable criteria for scores for TBAL are closely related to the ordinal ranking criterion [17]. To get such scores, Corbière et al. [6] add a module in the net for failure prediction, Zhu et al. [64] switch to sharpness aware minimization [7] to learn the model; CRL [35] regularizes the loss.

## 6 Conclusion

We studied issues with confidence scoring functions used in threshold-based auto-labeling (TBAL). We showed that the commonly used confidence functions and calibration methods can often be a bottleneck, leading to poor performance. We proposed `Colander` to learn confidence functions that are aligned with the TBAL objective. We evaluated our method extensively against common baselines on several real-world datasets and found that it improves the performance of TBAL significantly in comparison to the several common choices of confidence function. Our method is compatible with several choices of methods used for training the classifier in TBAL and using it in conjunction with them improves TBAL performance further. A limitation of `Colander` is that, similar to other post-hoc methods it also requires validation data to learn the confidence function. Reducing (or eliminating) this dependence on validation data could be an interesting future work.

## 7 Acknowledgments

This work was partly supported by funding from the American Family Data Science Institute. We thank Heguang Lin, Changho Shin, Dyah Adila, Tzu-Heng Huang, John Cooper, Aniket Rege, Daiwei Chen and Albert Ge for their valuable inputs. We thank the anonymous reviewers for their valuable comments and constructive feedback on our work.

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

# Supplementary Material Organization

The supplementary material is organized as follows. We provide deferred details of the background and motivation section in Appendix A of the method in Appendix B. Then, in Appendix C, we provide additional experimental results and details of the experiment protocol and hyperparameters used for the experiments. Our code with instructions to run, is uploaded along with the paper.

# A    Appendix to the Background and Motivation Section

## A.1    Detailed comparison with active learning and self-training

To illustrate the differences between TBAL and the combination of active learning (AL) and self-training for the task of data labeling, we run an experiment on the 2 concentric circles data setting as used in [56]. The details are as follows:

**Data setting.** We generate two concentric circles with points in the outer circle belonging to one class and the inner circle belonging to the other class. The total number of points generated is 10,000 of which we use 2000 for validation.

**Methods.** We run TBAL, AL+Self-Training, and AL+Self-Training+SC, using logistic regression. The combination of AL+Self-Training means, in each iteration, the algorithm queries human-labeled data points and pseudo-labels the points in the unlabeled data using self-training and adds both the human-labeled and pseudo-labeled points in the training pool. With this procedure, AL+Self-Training first learns the best classifier ($\hat{h}_{\mathrm{al-st}}$) with the given budget of maximum training points ($N_t$) that can be

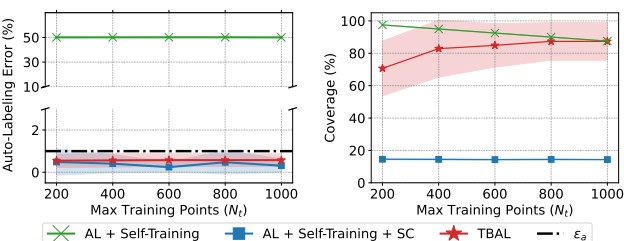

Figure 5: Results of experiment on 2-concentric circles to show the differences between TBAL, AL and ST.

queried from humans. Then it auto-labels all the remaining unlabeled points with this classifier's predictions. For AL+Self-Training+SC, we do selective auto-labeling using $\hat{h}_{\mathrm{al-st}}$, i.e., only auto-label the points where the classifier will have an error at most $\epsilon_a$. We use $\epsilon_a = 1\%$ here.

**Results and discussion.** The Figure 5 shows auto-labeling error and coverage achieved by these methods when run with different choices of human-labeled data budget for training. First, we can see that even with linear classifiers TBAL is able to auto-label a huge chunk of the data (high coverage) while maintaining auto-labeling error below the tolerance level of 1% On the other hand, methods like AL+Self-Training (+SC) that try to first learn the optimal classifier in the given function class either have high auto-labeling error or very low coverage. These results are also consistent with the observations in [56] on the comparison between TBAL and AL, AL+SC. While such findings confirm the notion that there are differences—and, at least in some settings, advantages—for the TBAL approach compared to other techniques, we reiterate that our goal is to understand and improve the role of the confidence function within TBAL, rather than comparing TBAL to other techniques.

## A.2    Details of the motivating experiment in section 2

We run TBAL for a single round on the CIFAR-10 dataset with a SimpleCNN classification model with around 5.8M parameters [20]. We randomly sampled 4,000 points for training the classifier and randomly sampled 1,000 points as validation data. We train the model to zero training error using minibatch SGD with learning rate 1e-3, weight decay 1e-3 [15, 25], momentum 0.9, and batch size 32. The trained model has validation accuracy around 55%, implying we could hope to get coverage around 55%. We run the auto-labeling procedure with an error tolerance of 5%.

# B  Additional Details on the Method

## B.1  Detailed algorithms

See Algorithms 1, 2 and 3.

## B.2  Tightness of surrogates.

The surrogate auto-labeling error and coverage introduced to relax the optimization problem (P2) is indeed a good approximation of the actual auto-labeling error and coverage. To see this, we use a toy data setting of $x \sim \mathrm{Uniform}(0, 1)$ with $1-$dimensional threshold classifier $h_\theta(x) = \mathbb{1}(x \geq \theta)$. For any $x$, let true labels $y = h_{0.5}(x)$ and consider the confidence function $g_w(x) = |w - x|$. Let $\hat{y} = h_{0.25}(x)$ and consider the points on the side where $\hat{y} = 1$. We plot actual and surrogate errors in Figure 6(a) and the surrogate and actual coverage in Figure 6(a).

for three choices of $\alpha$. As expected, the gap between the surrogates and the actual functions diminishes as we increase the $\alpha$.

## B.3  Active querying strategy.

We employ the margin-random query approach to select the next batch of training data. This method involves sorting points based on their margin (uncertainty) scores and selecting the top $Cn_b$ points, from which $n_b$ points are randomly chosen. This strategy provides a straightforward and computationally efficient way to balance the exploration-exploitation trade-off. It's important to acknowledge the existence of alternative active-querying strategies; however, we adopt the margin-random approach as our standard to maintain a focus on evaluating various choices of confidence functions for auto-labeling. Note that while we use the new confidence

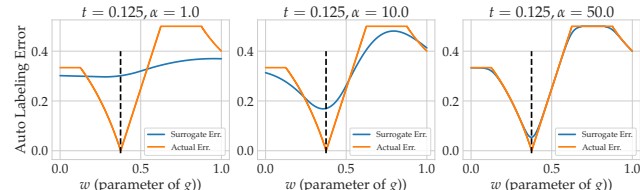

(a) Auto-labeling error and surrogate error at various $\alpha$.

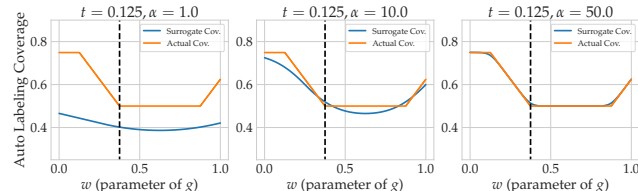

(b) Auto-labeling coverage and surrogate coverage at various $\alpha$.

Figure 6: Illustration of the tightness of surrogate error and coverage functions based on the choice of $\alpha$.

scores computed using post-hoc methods for auto-labeling, we do not use these scores in active querying. Instead, we use the softmax scores from the model for this. We do this to avoid conflating the study with the study of active querying strategies. We use $C = 2$ for all experiments.

## B.4 Glossary

The notation is summarized in Table 3 below.

| Symbol | Definition |
|---|---|
| $\mathbb{1}(E)$ | indicator function of event $E$. It is 1 if $E$ happens and 0 otherwise. |
| $\mathcal{X}$ | feature space. |
| $\mathcal{Y}$ | label space i.e. $1, 2, \ldots k$. |
| $\mathcal{H}$ | hypothesis space (model class for the classifiers). |
| $\mathcal{G}$ | class of confidence functions. |
| $k$ | number of classes. |
| $\mathbf{x}, y$ | $\mathbf{x}$ is an element in $\mathcal{X}$ and $y$ is its true label. |
| $f^*$ | unknown groundtruth labeling function. |
| $h$ | a hypothesis (model) in $\mathcal{H}$. |
| $g$ | confidence function $g : \mathcal{X} \to \Delta^k$. |
| $\epsilon_a$ | auto-labeling error tolerance. |
| $X_u$ | given pool of unlabeled data points. |
| $X_u^{(i)}$ | unlabeled data left at the beginning of $i$th round. |
| $A$ | indices of points that are auto-labeled. |
| $X_u(A)$ | subset of points in $X_u$ with indices in $A$, i.e. the set of auto-labeled points. |
| $\hat{h}^{(i)}$ | ERM solution and auto-labeling thresholds respectively in $i$th round. |
| $\mathbf{t}$ | $k$ dimensional vector of thresholds. |
| $\mathbf{t}[y]$ | $y$th entry of $\mathbf{t}$ i.e. the threshold for class $y$. |
| $g(\mathbf{x})[y]$ | the confidence score for class $y$ output by confidence function $g$ on data point $\mathbf{x}$. |
| $N_u$ | number of unlabeled points, i.e. size of $X_u$. |
| $N_t$ | number of manually labeled points that can be used for training $h$. |
| $N_a$ | Total auto-labeled points in $D_{\text{out}}$. |
| $\nu$ | fraction of $D_{\text{val}}$ that can be used for training post-hoc calibrator. |
| $\tilde{y}_i$ | label assigned to the $i$th point by the algorithm. It could be either $y_i$ or $\hat{y}_i$. |
| $y_i$ | groundtruth label for the $i$th point. |
| $\hat{y}_i$ | predicted label for the $i$th point by classifier. |
| $D_{\text{query}}^{(i)}$ | labeled data queried from oracle (human) in the $i$th round. |
| $D_{\text{train}}^{(i)}$ | training data to learn $\hat{h}^{(i)}$ in the $i$th round. |
| $D_{\text{val}}^{(i)}$ | validation data in the $i$th round. |
| $D_{\text{cal}}^{(i)}$ | calibration data in the $i$th round to learn a post-hoc $g$. |
| $D_{\text{th}}^{(i)}$ | part of validation data in the $i$th round to estimate threshold $\mathbf{t}$. |
| $D_{\text{auto}}^{(i)}$ | part of $X_u^{(i)}$ that got auto-labeled in the $i$th round. |
| $D_{\text{out}}$ | Output labeled data, including auto-labeled and human labeled data. |
| $\mathcal{E}(g, \mathbf{t} \mid h)$ | population level auto-labeling error, see eq. (2). |
| $\mathcal{P}(g, \mathbf{t} \mid h)$ | population level auto-labeling coverage, see eq. (2). |
| $\widehat{\mathcal{E}}(g, \mathbf{t} \mid h, D)$ | estimated auto-labeling error, see eq. (4). |
| $\widehat{\mathcal{P}}(g, \mathbf{t} \mid h, D)$ | estimated auto-labeling coverage, see eq. (3). |
| $\widetilde{\mathcal{E}}(g, \mathbf{t} \mid h, D)$ | surrogate estimated auto-labeling error, see eq. (6). |
| $\widetilde{\mathcal{P}}(g, \mathbf{t} \mid h, D)$ | surrogate estimated auto-labeling coverage, see eq. (5). |

Table 3: Glossary of variables and symbols used in this paper.

**Algorithm 1** Threshold-based Auto-Labeling (TBAL)

---

**Input:** Unlabeled data $X_u$, labeled validation data $D_{\text{val}}$, auto labeling error tolerance $\epsilon_a$, $N_t$ training data query budget, seed data size $n_s$, batch size for active query $n_b$, calibration data fraction $\nu$, space of thresholds $T$, coverage lower bound $\rho_0$, label space $\mathcal{Y}$.

**Output:** Auto-labeled dataset $D_{\text{out}}$.

1: **procedure** TBAL($X_u, D_{\text{val}}, \epsilon_a, N_t, n_s, n_b, \nu, \rho_0, T, \mathcal{Y}$)
2:    ▷ /*** Initialization. ***/
3:    $D_{\text{query}}^{(1)} \leftarrow$ RANDOMQUERY($X_u, n_s$).  ▷ Randomly select $n_s$ points and get human labels.
4:    $X_u^{(1)} \leftarrow X_u \setminus \{\mathbf{x} : (\mathbf{x}, y) \in D_{\text{query}}^{(1)}\}$.        ▷Remove these points from the unlabeled pool.
5:    $D_{\text{val}}^{(1)} \leftarrow D_{\text{val}}; D_{\text{train}}^{(0)} \leftarrow \emptyset$.        ▷Validation data for the first round is full $D_{\text{val}}$.
6:    $D_{\text{out}} \leftarrow D_{\text{query}}^{(1)}; n_t^{(1)} \leftarrow n_s; i \leftarrow 1$.        ▷Add human-labeled data to the output $D_{\text{out}}$.

7:    ▷ /*** Run the auto-labeling loop. ***/
8:    ▷ /* Until no more unlabeled points are left or the budget for training data is exhausted. */

9:    **while** $X_u^{(i)} \neq \emptyset$ and $n_t^{(i)} \leq N_t$ **do**
10:        $D_{\text{train}}^{(i)} \quad \leftarrow D_{\text{train}}^{(i-1)} \cup D_{\text{query}}^{(i)}$.  ▷Include human-labeled points in the training data.
11:        $\hat{h}_i \qquad \leftarrow$ TRAINMODEL($\mathcal{H}, D_{\text{train}}^{(i)}$).        ▷Train a classification model.
12:        $D_{\text{cal}}^{(i)}, D_{\text{th}}^{(i)} \leftarrow$ RANDOMSPLIT($D_{\text{val}}^{(i)}, \nu$).        ▷Randomly split current validation data.

13:        ▷ /*** Colander block, to learn the new confidence function $\hat{g}_i$. ***/

14:        $\hat{g}_i, \hat{\mathbf{t}}_i' \quad \leftarrow \arg\min_{g \in \mathcal{G}, \mathbf{t} \in T^k} -\widetilde{\mathscr{P}}(g, \mathbf{t} \mid \hat{h}_i, D_{\text{cal}}^{(i)}) + \lambda \widetilde{\mathscr{E}}(g, \mathbf{t} \mid \hat{h}_i, D_{\text{cal}}^{(i)})$.     ▷ Colander.

15:        ▷ /*** Estimate auto-labeling thresholds using $\hat{g}_i$ and $D_{\text{th}}^{(i)}$. See Algorithm 2. ***/

16:        $\hat{\mathbf{t}}_i \qquad \leftarrow$ ESTTHRESHOLD($\hat{g}_i, \hat{h}_i, D_{\text{th}}^{(i)}, \epsilon_a, \rho_0, T, \mathcal{Y}$).
17:        ▷ /*** Auto-label the points having scores above the thresholds. ***/
18:        $\widetilde{D}_u^{(i)} \quad \leftarrow \{(\mathbf{x}, \hat{h}_i(\mathbf{x})) : \mathbf{x} \in X_u^{(i)}\}$.
19:        $D_{\text{auto}}^{(i)} \leftarrow \{(\mathbf{x}, \hat{y}) \in \tilde{D}_u^{(i)} : \hat{g}_i(\mathbf{x})[\hat{y}] \geq \hat{t}_i[\hat{y}]\}$.
20:        $X_u^{(i)} \quad \leftarrow X_u^{(i)} \setminus \{\mathbf{x} : (\mathbf{x}, \hat{y}) \in D_{\text{auto}}^{(i)}\}$. ▷Remove auto-labeled points from unlabeled set.

21:        $\widetilde{D}_{\text{val}}^{(i)} \quad \leftarrow \{(\mathbf{x}, \hat{h}_i(\mathbf{x})) : (\mathbf{x}, y) \in D_{\text{val}}^{(i)}\}$.
22:        $D_{\text{val}}^{(i+1)} \leftarrow \{(\mathbf{x}, \hat{y}) \in \tilde{D}_{\text{val}}^{(i)} : \hat{g}_i(\mathbf{x})[\hat{y}] < \hat{t}_i[\hat{y}]\}$.        ▷Remove validation points from the auto-labeling region.

23:        ▷ /*** Get the next batch of manually labeled data using an active querying strategy. ***/

24:        $D_{\text{query}}^{(i+1)} \leftarrow$ ACTIVEQUERY($\hat{h}_i, X_u^{(i)}, n_b$).
25:        $X_u^{(i+1)} \leftarrow X_u^{(i)} \setminus \{\mathbf{x} : (\mathbf{x}, y) \in D_{\text{query}}^{(i+1)}\}$.        ▷Remove human-labeled data from the unlabeled pool.
26:        $D_{\text{out}} \quad \leftarrow D_{\text{out}} \cup D_{\text{auto}}^{(i)} \cup D_{\text{query}}^{(i+1)}$. ▷Add the auto-labeled and manually labeled points in the output data.
27:        $n_t^{(i+1)} \leftarrow n_t^{(i)} + n_b$.
28:        $i \quad \leftarrow i + 1$.
29:    **end while**
30:    **return** $D_{\text{out}}$.
31: **end procedure**

**Algorithm 2** Estimate Auto-Labeling Threshold

**Input:** Confidence function $\hat{g}_i$, classifier $\hat{h}_i$, Part of validation data $D_{\text{th}}^{(i)}$ for threshold estimation, auto labeling error tolerance $\epsilon_a$, space of thresholds $T$, coverage lower bound $\rho_0$, label space $\mathcal{Y}$.
**Output:** Auto-labeling thresholds $\hat{\mathbf{t}}_i$, where $\hat{\mathbf{t}}_i[y]$ is the threshold for class $y$.

1: **procedure** ESTTHRESHOLD($\hat{g}_i, \hat{h}_i, D_{\text{th}}^{(i)}, \epsilon_a, \rho_0, T, \mathcal{Y}$)
2:  ▷ /\*\*\* Estimate thresholds for each class. \*\*\*/
3:  **for** $y \in \mathcal{Y}$ **do**
4:   $D_{\text{th}}^{(i,y)} \leftarrow \{(\mathbf{x}', y') \in D_{\text{th}}^{(i)} : y' = y\}$. ▷Group points class-wise.
5:   ▷ /\*\*\* Only evaluate thresholds with est. coverage at least $\rho_0$. \*\*\*/
6:   $T_y' \leftarrow \{t \in T : \widehat{\mathcal{P}}(\hat{g}_i, t \mid \hat{h}_i, D_{\text{th}}^{(i,y)}) \geq \rho_0\} \cup \{\infty\}$.
7:   ▷ /\*\*\* Estimate auto-labeling error at each threshold. Pick the smallest threshold with the sum of estimated error and $C_1$ times the std. deviation is below $\epsilon_a$. $C_1$ is set to 0.25 here. \*\*\*/
8:   $\hat{\mathbf{t}}_i[y] \leftarrow \min\{t \in T_y' : \widehat{\mathcal{E}}_y(\hat{g}_i, t|\hat{h}_i, D_{\text{th}}^{(i,y)}) + C_1\widehat{\zeta}(\widehat{\mathcal{E}}_y(\hat{g}_i, t|\hat{h}_i, D_{\text{th}}^{(i,y)})) \leq \epsilon_a\}$.
9:  **end for**
10:  **return** $\hat{\mathbf{t}}_i$.
11: **end procedure**

---

**Algorithm 3** Active Querying Strategy to Acquire Human-labeled Samples for Training

**Input:** Classifier $\hat{h}_i$, unlabeled data $X_u^{(i)}$, batch size $n_b$, constant $C \geq 1$.
**Output:** $D_{\text{query}}^{(i+1)}$, a subset of $X_u^{(i)}$ of size at most $n_b$ with human (groundtruth) labels.

1: **procedure** ACTIVEQUERY($\hat{h}_i, X_u^{(i)}, n_b$)
2:  $S_u^{(i)} \leftarrow$ Softmax scores from $\hat{h}_i$ for all points in $X_u^{(i)}$.
3:  $X' \leftarrow$ Top $C \times n_b$ points from $X_u^{(i)}$ sorted in ascending order on the scores $S_u^{(i)}$.
4:  $D_{\text{query}}^{(i+1)} \leftarrow$ RANDOMQUERY($X', n_b$).
5:  **return** $D_{\text{query}}^{(i+1)}$.
6: **end procedure**

---

**Algorithm 4** Select a Subset of Points Randomly and Obtain Human Labels

**Input:** $X, n$.
**Output:** $D$, a subset of $X$ of size at most $n$ with human (groundtruth) labels.

1: **procedure** RANDOMQUERY($X, n$)
2:  **if then**$|X| > n$
3:   $X'' \leftarrow$ randomly select $n$ points from $X$.
4:  **else**
5:   $X'' \leftarrow X$.
6:  **end if**
7:  $D \leftarrow \{(\mathbf{x}, \texttt{human\_label}(\mathbf{x}) : \mathbf{x} \in X''\}$.
8:  **return** $D$.
9: **end procedure**

# C  Additional Experiments and Details

## C.1  Experiments on $N_t$, $N_v$ and $\nu$

We need to understand the effect of training data query budget i.e. $N_t$, the total validation data $N_v$, and the data that can be used for calibrating the model i.e. the calibration data fraction $\nu$ on the auto-labeling objective. As varying these hyperparameters on each train-time method is expensive, we experimented with only Squentropy as it was the best-performing method across settings for various datasets.

When we vary the budget for training data $N_t$, we observe from Figure 7 that our method does not require a lot of data to train the base model, i.e. achieving low auto-labeling error and high coverage with a low budget. While other methods benefit from having more training data for auto-labeling objectives, it comes at the expense of reducing the available data for validation.

From Figure 8, we observe that, while the coverage of our method remains the same across different $N_v$, it reduces for other methods. The cause of this phenomenon can be attributed to the fact that we are borrowing the data from the training budget as it limits the performance of the base model, which in turn limits the auto-labeling objective.

As we increase the percentage of data that can be used to calibrate the model, i.e., $\nu$, we note from Figure 9 that other methods improve the coverage, which can be understood from the fact that when more data is available for calibrating the model, the model becomes better in terms of the auto-labeling objective. But it's interesting to note that even with a low calibration fraction, our method achieves superior coverage compared to other methods. It is also important to note that the auto-labeling error increases as we increase $\nu$. This is because when $\nu$ increases, the number of data points used to estimate the threshold decreases, leading to a less granular and precise threshold.

| Feature | Model | Error | Coverage |
|---------|-------|-------|----------|
| Pre-logits | Two Layer | $4.6 \pm 0.3$ | $82.8 \pm 0.5$ |
| Logits | Two Layer | $3.2 \pm 1.3$ | $82.8 \pm 0.3$ |
| Concat | Two Layer | $3.3 \pm 0.8$ | $82.9 \pm 0.4$ |

Table 4: Auto-labeling error and coverage for the 3 feature representations we could use for 20 Newsgroup. As we can see, the feature representation does not lead to a significant difference in auto-labeling error and coverage.

| Feature | Model | Error | Coverage |
|---------|-------|-------|----------|
| Pre-logits | Two Layer | $2.1 \pm 0.5$ | $79.0 \pm 0.2$ |
| Logits | Two Layer | $3.1 \pm 0.4$ | $76.5 \pm 0.9$ |
| Concat | Two Layer | $2.3 \pm 0.5$ | $79.0 \pm 0.3$ |

Table 5: Auto-labeling error and coverage for the 3 feature representations we could use for CIFAR10 SimpleCNN. As we can see, the feature representation does not lead to a significant difference in auto-labeling error and coverage.

## C.2  Experiments on `Colander` input

Figure 4 illustrates that we could use logits (last layer's representations), pre-logits (second last layer's representations), or the concatenation of these two as the input to $g$. To help us decide which one we should use, we conduct a hyperparameter search for input features on the CIFAR-10 and 20 Newsgroup dataset using the Squentropy train-time method. Table 4 and 5 present the auto-labeling error and coverage of using the 3 types of feature representations. As we can see, all feature representation leads to a similar auto-labeling error and coverage, and in some cases, it is better to include pre-logits as well. Thus, we use concatenated representation (Concat), for more flexibility.

## C.3 Experiments on $\epsilon_a$

We run TBAL with five values of $\epsilon_a \in \{0.01, 0.025, 0.05, 0.075, 0.1\}$ and report the results in Table 6. As expected the auto-labeling error is high with larger values of and smaller with small $\epsilon_a$.

| Post-hoc | $\epsilon_a = 0.01$ | | $\epsilon_a = 0.025$ | | $\epsilon_a = 0.05$ | | $\epsilon_a = 0.075$ | | $\epsilon_a = 0.1$ | |
|---|---|---|---|---|---|---|---|---|---|---|
| | Err (↓) | Cov (↑) | Err (↓) | Cov (↑) | Err (↓) | Cov (↑) | Err (↓) | Cov (↑) | Err (↓) | Cov (↑) |
| Softmax | 5.86 ± 0.38 | 12.73 ± 1.61 | 5.86 ± 0.38 | 12.73 ± 1.61 | 4.78 ± 0.21 | 14.01 ± 2.08 | 6.80 ± 0.47 | 16.73 ± 1.19 | 9.03 ± 0.17 | 21.28 ± 0.82 |
| TS | 8.19 ± 0.88 | 19.44 ± 1.16 | 8.19 ± 0.88 | 19.44 ± 1.16 | 7.26 ± 0.29 | 23.15 ± 0.7 | 9.24 ± 0.78 | 22.49 ± 0.74 | 11.63 ± 0.51 | 25.79 ± 1.97 |
| Dirichlet | 8.22 ± 0.4 | 16.94 ± 1.2 | 8.22 ± 0.4 | 16.94 ± 1.2 | 7.6 ± 0.48 | 22.36 ± 1.18 | 9.68 ± 0.82 | 18.65 ± 0.97 | 11.26 ± 1.16 | 24.91 ± 2.09 |
| SB | 6.15 ± 0.52 | 11.74 ± 0.57 | 6.15 ± 0.52 | 11.74 ± 0.57 | 6.09 ± 0.35 | 18.58 ± 1.13 | 7.81 ± 0.65 | 17.37 ± 1.3 | 9.13 ± 1.08 | 20.52 ± 1.11 |
| Top-HB | 5.76 ± 0.42 | 9.89 ± 0.55 | 5.76 ± 0.42 | 9.89 ± 0.55 | 5.95 ± 0.7 | 15.58 ± 1.92 | 7.45 ± 0.8 | 13.84 ± 0.78 | 8.71 ± 1.37 | 17.9 ± 0.56 |
| **Ours** | **1.2 ± 0.18** | **78.33 ± 0.76** | **1.32 ± 0.21** | **78.75 ± 0.4** | **2.96 ± 0.2** | **78.48 ± 0.17** | **4.3 ± 0.23** | **78.94 ± 0.42** | **6.29 ± 0.5** | **78.97 ± 0.46** |

Table 6: $\epsilon_a$ **variation.** Dataset: CIFAR-10, Train-time method: Vanilla.

## C.4 Single vs multi-round TBAL

We further demonstrate that the performance gains are due to the use of `Colander`, even if methods use multiple rounds. To do so, we show the evolution of coverage and error over multiple rounds in Figure 16. The effects of using `Colander` are visible from the first round itself, and the following rounds improve performance further. We also run a single round (passive) variant of TBAL where we sample all the human-labeled points for

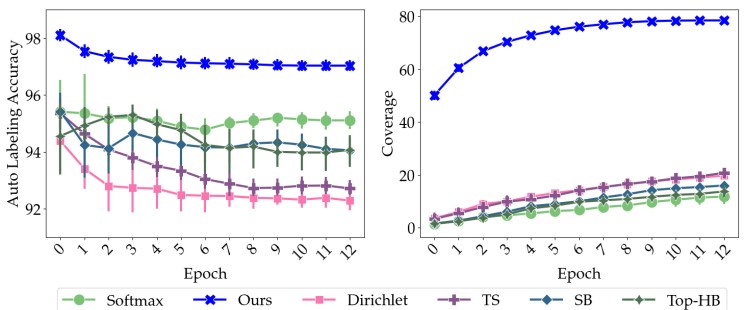

Figure 16: Per-epoch metrics for all post-hoc methods for CIFAR10 setting. (left) Auto-labeling accuracy (right) Coverage. Train-time method is vanilla.

training ($N_t$) randomly at once, train a classifier, do auto-labeling, and then stop. This setting avoids confounding due to multiple rounds. We observe that using `Colander` yields significantly higher coverage in comparison to the baselines (see Table 7). This reinforces the fact that the gains in the multi-round TBAL are directly due to `Colander`, while multiple rounds of data selection, training, and auto-labeling are superior to this single round version.

## C.5 Experiments on different architectures

In TBAL it is not a priori clear what model the practitioner should use. The overall system is flexible enough to work with any chosen model class. Our focus is on evaluating the effect of various training time and post-hoc methods designed to improve the confidence functions for any given model. To answer the query, we ran experiments with Resnet18 and ViT models in the CIFAR-10 setting (see Table 8). As we expected there are variations in the results in the baselines due to model choices but our method maintains high performance irrespective of the classification model used. This is due to its ability to learn confidence scores tailored for TBAL.

| Post-hoc | Err (↓) | Cov (↑) |
|---|---|---|
| Softmax | 2.7 ± 0.54 | 11.06 ± 1.46 |
| TS | 3.04 ± 0.49 | 12.03 ± 1.98 |
| Dirichlet | 2.98 ± 0.32 | 11.22 ± 2.1 |
| SB | 2.72 ± 0.34 | 9.75 ± 1.33 |
| Top-HB | 1.83 ± 0.61 | 5.50 ± 1.08 |
| **Ours** | **2.02 ± 0.28** | **49.62 ± 0.69** |

Table 7: Results with single round of auto-labeling. Dataset and model: CIFAR-10 setting in the paper.

| Post-hoc | Err (↓) | Cov (↑) |
|---|---|---|
| Softmax | 14.02 ± 1.83 | 2.03 ± 0.31 |
| TS | 19.32 ± 2.51 | 2.54 ± 0.33 |
| Dirichlet | 17.27 ± 3.26 | 2.87 ± 0.55 |
| SB | 9.22 ± 10.91 | 0.46 ± 0.51 |
| Top-HB | 0.00 ± 0.00 | 0.00 ± 0.00 |
| **Ours** | **2.62 ± 0.32** | **75.56 ± 0.15** |

| Post-hoc | Err (↓) | Cov (↑) |
|---|---|---|
| Softmax | 4.48 ± 0.23 | 33.24 ± 1.14 |
| TS | 6.38 ± 0.47 | 39.14 ± 1.96 |
| Dirichlet | 6.30 ± 0.41 | 37.99 ± 1.47 |
| SB | 5.16 ± 0.23 | 35.32 ± 1.36 |
| Top-HB | 4.46 ± 0.40 | 29.66 ± 0.74 |
| **Ours** | **2.85 ± 0.25** | **78.56 ± 0.54** |

Table 8: Model variation. CIFAR-10 dataset with ViT (Left) and ResNet18 (Right), Train-time method Vanilla.

| Method | Hyperparameter | Values |
|---|---|---|
| Common | optimizer | SGD |
| | learning rate | 0.001, 0.01, 0.1 |
| | batch size | 32, 256 |
| | max epoch | 50, 100 |
| | weight decay | 0.001, 0.01, 0.1 |
| | momentum | 0.9 |
| CRL | rank target | softmax |
| | rank weight | 0.7, 0.8, 0.9 |
| FMFP | optimizer | SAM |

Table 9: Hyperparameters swept over for train-time methods. Those listed next to Common are the hyperparameters for the four train-time methods: Vanilla, CRL, FMFP, and Squentropy. Therefore, we do not list those again for each method. Note that for FMFP, we used SAM optimizer instead of SGD. For each method, we swept through all possible combinations of the possible values for each hyperparameter. Underlined values are only used on TinyImageNet since it is a complicated dataset containing 200 classes.

## C.6 Hyperparameters

The hyperparameters and their values we swept over are
listed in Table 9 and 10 for train-time and post-hoc methods, respectively.

## C.7 Train-time and post-hoc methods

### C.7.1 Train-time methods

1. *Vanilla*: Neural networks are commonly trained by minimizing the cross entropy loss using stochastic gradient descent (SGD) with momentum [1, 4]. We refer to this as the Vanilla training method. We also include weight decay to mitigate the overconfidence issue associated with this method [12].

2. *Squentropy* [19]: This method adds the average square loss over the incorrect classes to the cross-entropy loss. This simple modification to the Vanilla method leads to the end model with better test accuracy and calibration.

3. *Correctness Ranking Loss (CRL)* [35]: This method includes a term in the loss function of the vanilla training method so that the confidence scores of the model are aligned with the ordinal rankings criterion [17, 6]. The confidence functions satisfying this criterion produce high scores on points where the probability of correctness is high and low scores on points with low probabilities of being correct.

4. *FMFP* [64] aims to align confidence scores with the ordinal rankings criterion. It uses Sharpness Aware Minimizer (SAM) [7] to train the model, with the expectation that the flat minima would benefit the ordinal rankings objective of the confidence function.

### C.7.2 Post-hoc methods

1. *Temperature scaling* [12]: This is a variant of Platt scaling [12], a classic and one of the easiest parametric methods for post-hoc calibration. It rescales the logits by a learnable scalar parameter and has been shown to work well for neural networks.

2. *Top-Label Histogram-Binning* [14]: Since TBAL assigns the top labels (predicted labels) to the selected unlabeled points, it is appealing to only calibrate the scores of the predicted label. Building upon a rich line of histogram-binning methods (non-parametric) for post-hoc calibration [62], this method focuses on calibrating the scores of predicted labels.

3. *Scaling-Binning* [28]: This method combines parametric and non-parametric methods. It first applies temperature scaling and then bins the confidence function values to ensure calibration.

4. *Dirichlet Calibration* [26]: This method models the distribution of predicted probability vectors separately on instances of each class and assumes the class conditional distributions are Dirichlet distributions with different parameters. It uses linear parameterization for the distributions, which allows easy implementation in neural networks as additional layers and softmax output.

**Note:** For binning methods, uniform mass binning [62] has been a better choice over uniform width binning. Hence, we use uniform mass binning as well.

### C.8 Compute resources and time

Our experiments were conducted on machines equipped with the NVIDIA RTX A6000 and NVIDIA GeForce RTX 4090 GPUs. The wall clock time of our method is similar to other post-hoc methods. For instance, a single run on the CIFAR-10 setting on NVIDIA RTX A6000 takes around 1.5 hours with post-hoc methods and roughly 1 hour without post-hoc methods. The additional time taken by our method over the baselines not doing any post-hoc calibration is traded-off the by the quality and quantity of the auto-labeled data it outputs. We leave a thorough benchmarking of wall clock time and its optimization for future work.

### C.9 Detailed dataset and model

1. The MNIST dataset [30] consists of $28 \times 28$ grayscale images of hand-written digits across 10 classes. It was used alongside the LeNet5 [31], a convolutional neural network, for auto-labeling.

2. The CIFAR-10 dataset [24] contains $3 \times 32 \times 32$ color images across 10 classes. We utilized its raw pixel matrix in conjunction with SimpleCNN [20], a convolutional neural network with approximately 5.8M parameters, for auto-labeling.

3. Tiny-ImageNet [29] is a color image dataset that consists of 100K images across 200 classes. Instead of using the $3 \times 64 \times 64$ raw pixel matrices as input, we utilized CLIP [43] to derive embeddings within the $\mathbb{R}^{512}$ vector space. We used a 3-layer perceptron (1,000-500-300) as the auto-labeling model.

4. 20 Newsgroups [34, 40] is a natural language dataset comprising around 18,000 news posts across 20 topics. We used the FlagEmbedding [58] to map the textual data into $\mathbb{R}^{1024}$ embeddings. We used a 3-layer perceptron (1,000-500-30) as the auto-labeling model.

### C.10 Detailed experiments protocol

We predefined TBAL hyperparameters for each dataset-model pair and the hyperparameters we will sweep for each train-time and post-hoc method in Table 9 and Table 10 respectively. For a dataset-model pair, initially, we perform a hyperparameter search for the train-time method. Subsequently, we optimize the hyperparameters for post-hoc methods while keeping the train-time method fixed with the previously found optimum hyperparameter for that dataset-model pair.

We fix the hyperparameters for the train-time method while searching hyperparameters for the post-hoc method to alleviate computational budget throttle. We effectively reduce the search space to the sum of the cardinalities of unique hyper-parameter combinations across the two methods instead of a larger multiplicative product. Furthermore, due to the independent nature of these hyper-parameter combinations, TBAL runs can be highly parallelized to expedite the search process.

Since TBAL operates iteratively to acquire human labels for model training, selecting hyper-parameters at each round of TBAL could quickly become intractable and lose its practical significance. To better align with its practical usage, we only conducted a hyperparameter search for the initial TBAL round. The specific set of hyperparameters used for the search are reported in Table 10.

After completing the hyperparameter search for train-time and post-hoc methods, the determined hyperparameter combinations are subjected to a full evaluation across all iterations of TBAL. At the end of each iteration, the auto-labeled points are evaluated against their ground truth labels to determine their auto-labeling error. These points are then added to the auto-labeled set, where their ratio to the total amount of unlabeled data determines the coverage. This iterative process continues until all unlabeled data are exhaustively labeled by either the oracle or through auto-labeling in the final iteration. The auto-labeling error and coverage at the final iteration of TBAL are then recorded.

Since TBAL incorporates randomized components as detailed in Algorithm 1, we ran the algorithm 5 times, each with a unique random seed while maintaining the same hyperparameter combination. We then recorded the results from the final iteration of these runs and calculated the mean and standard deviation of both auto-labeling error and coverage. These figures are reported in Table 2.

A limitation of the grid search approach in hyper-parameter optimization becomes apparent when our predefined hyper-parameter choices result in sub-optimal coverage and auto-labeling errors. Using these sub-optimal hyper-parameters can adversely affect the multi-round iterative process in TBAL, prompting the need for repetitive searches to find more effective hyper-parameters. When encountering such scenarios, TBAL users should explore additional hyper-parameter options until satisfactory performance is achieved in the initial round. However, we opted for a more straightforward approach to hyper-parameter selection, mindful of the computational demands of repeatedly optimizing multiple hyper-parameters across different methods. In scenarios expressed conditionally, we retained the top-1 hyper-parameter combination for any given method if it achieved the highest coverage while adhering to the specified error margin ($\epsilon_a$). If no hyper-parameter combinations yielded an auto-labeling error at most equal to the error margin ($\epsilon_a$), we then chose the hyper-parameter combination with the lowest auto-labeling error, regardless of its coverage. In the case of ties, we resolved them through random selection. This process results in obtaining singular values for each choice of hyper-parameter after completing each method's hyper-parameter search.

# D    Broader Impact

This paper contributes to the advancement of the practice of creating labeled datasets in machine learning. While our work has various possible societal implications, we do not identify any specific concerns that require special attention in this context.


Figure 7: Autolabeling error and coverage of different post-hoc methods on CIFAR-10 for various $N_t$

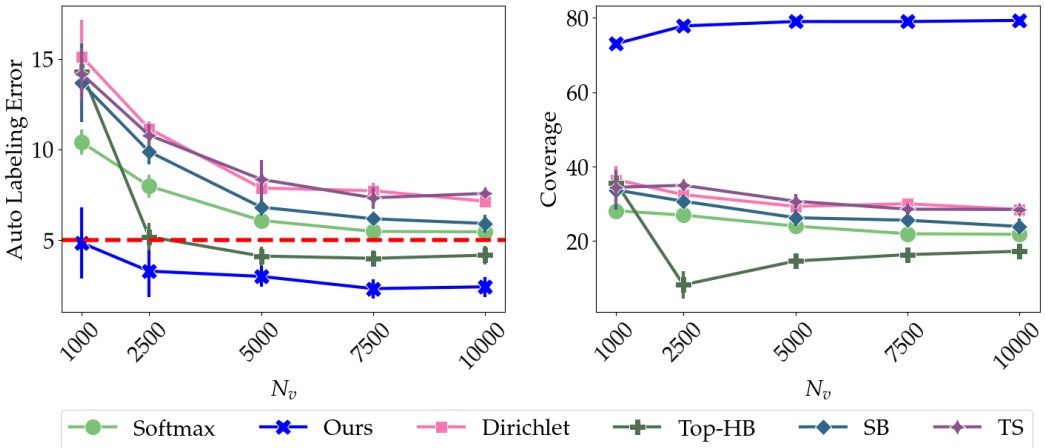

Figure 8: Autolabeling error and coverage of different post-hoc methods on CIFAR-10 for various $N_v$

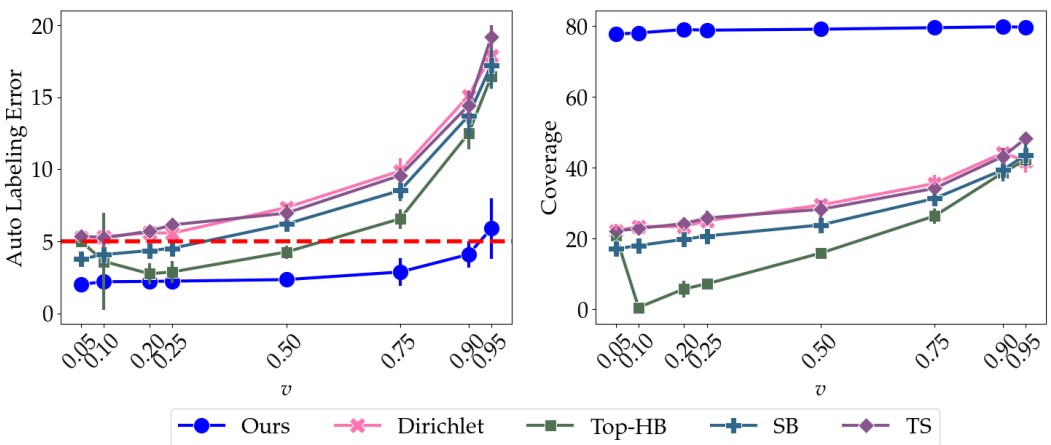

Figure 9: Autolabeling error and coverage of different post-hoc methods on CIFAR-10 for various $\nu$

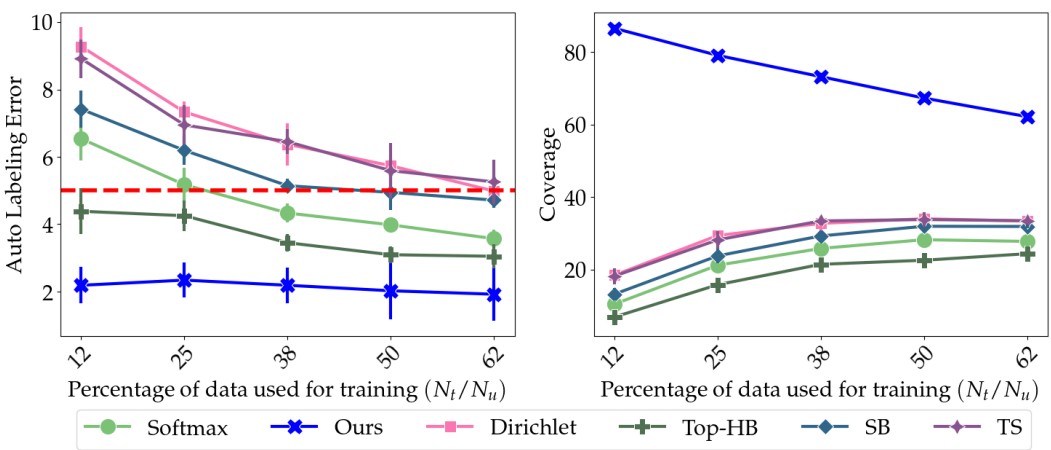

Figure 10: Auto-labeling error and coverage for different post-hoc methods on CIFAR-10 while we vary $N_t$. $N_u = 40,000$ is the size of the given unlabeled pool.

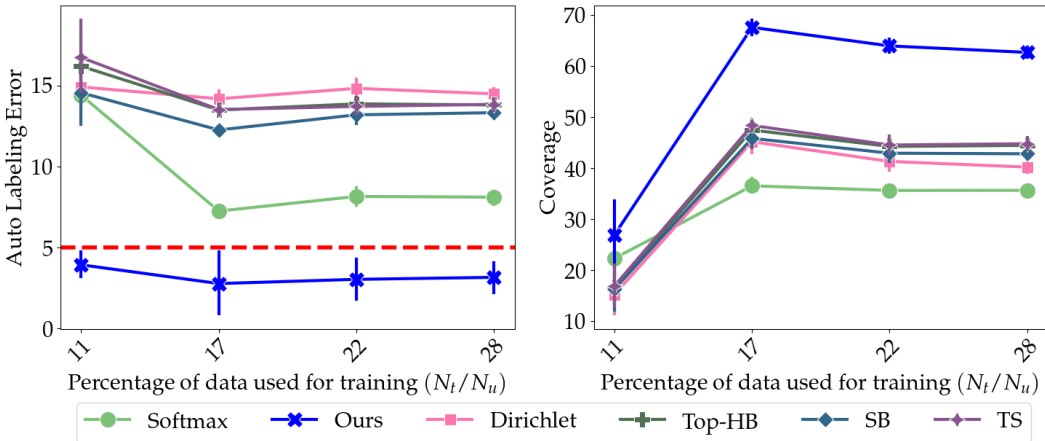

Figure 11: Auto-labeling error and coverage for different post-hoc methods on Tiny-ImageNet while we vary $N_t$. $N_u = 90,000$ is the size of the given unlabeled pool.

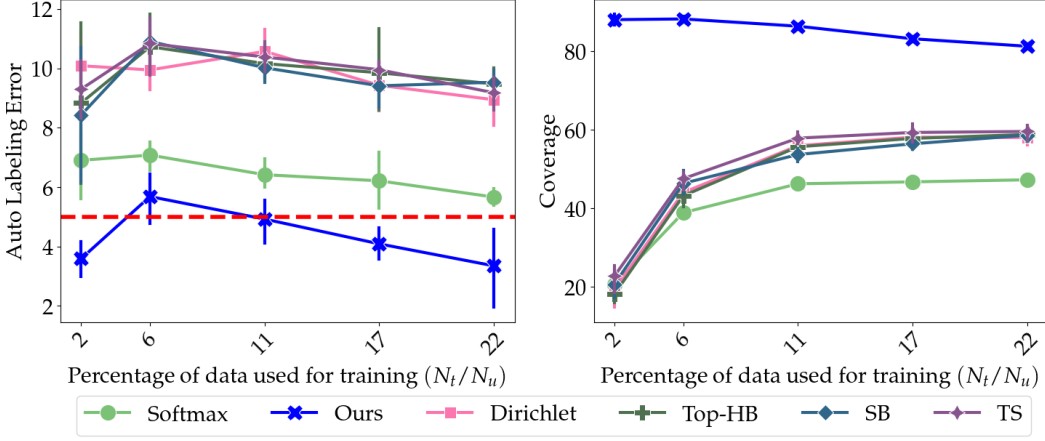

Figure 12: Auto-labeling error and coverage for different post-hoc methods on 20 Newsgroups while we vary $N_t$. $N_u = 9,052$ is the size of the given unlabeled pool.

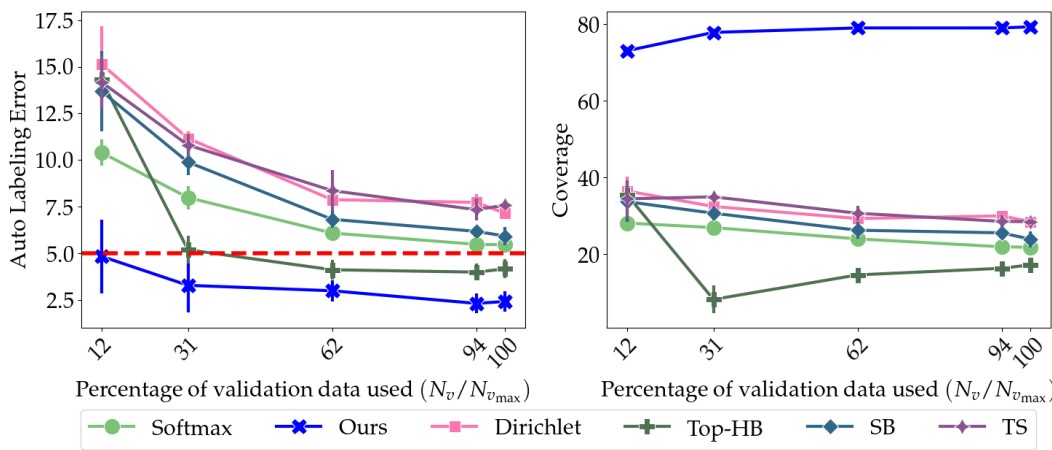

Figure 13: Auto-labeling error and coverage for different post-hoc methods on CIFAR-10 while we vary $N_v$. $N_{v_{\max}} = 8,000$ is the maximum number of points available for validation.

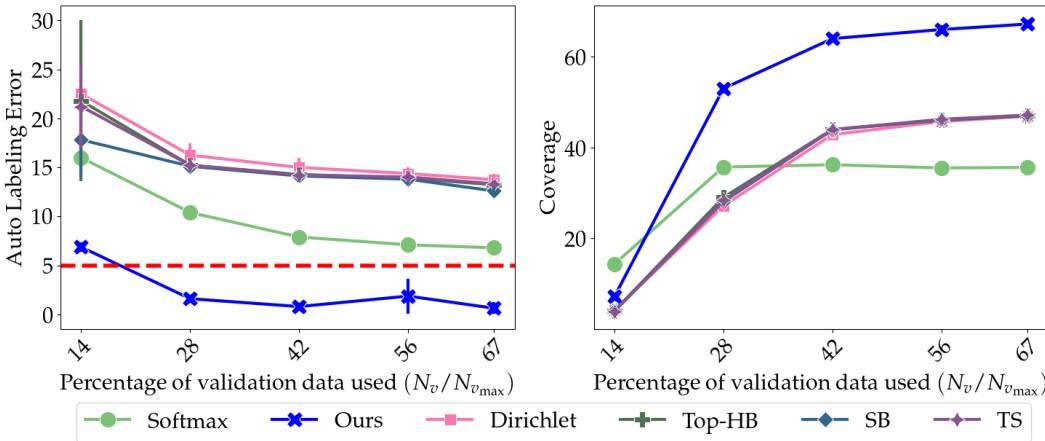

Figure 14: Auto-labeling error and coverage for different post-hoc methods on Tiny-ImageNet while we vary $N_v$. $N_{v_{\max}} = 18,000$ is the maximum number of points available for validation.

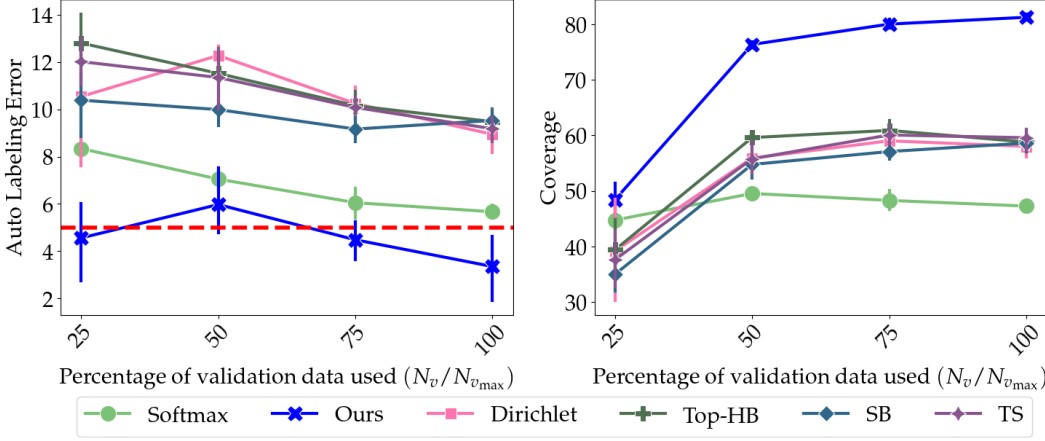

Figure 15: Auto-labeling error and coverage for different post-hoc methods on 20 Newsgroups while we vary $N_v$. $N_{v_{\max}} = 1,600$ is the maximum number of points available for validation.

| Method | Hyperparameter | Values |
|---|---|---|
| Temperature scaling | optimizer | Adam |
| | learning rate | 0.001, 0.01, 0.1 |
| | batch size | 64 |
| | max epoch | 500 |
| | weight decay | 0.01, 0.1, 1 |
| Top-label histogram binning | points per bin | 25, 50 |
| Scaling-binning | number of bins | 15, 25 |
| | learning rate | 0.001, 0.01, 0.1 |
| | batch size | 64 |
| | max epoch | 500 |
| | weight decay | 0.01, 0.1, 1 |
| Dirichlet calibration | regularization parameter | 0.001, 0.01, 0.1 |
| Ours | $\lambda$ | 10, 100 |
| | features key | concat |
| | class-wise | independent |
| | optimizer | Adam |
| | learning rate | 0.01, 0.1 |
| | max epoch | 500 |
| | weight decay | 0.01, 0.1, 1 |
| | batch size | 64 |
| | regularize | false |
| | $\alpha$ | 0.01, 0.1, 1 |

Table 10: Hyperparamters swept over for post-hoc methods. For each method, we swept through all possible combinations of the possible values for each hyperparameter.

