# OpenReview forum: "Pearls from Pebbles: Improved Confidence Functions for Auto-labeling"
_NeurIPS.cc/2024/Conference — NeurIPS 2024 poster_

### Official Review · Reviewer_ffq7 · 2024-07-11

**Soundness:** 3
**Presentation:** 3
**Contribution:** 3
**Rating:** 6
**Confidence:** 2

**Summary:**

This paper introduces a threshold-based auto-labeling (TBAL) method called Colander to maximize TBAL performance by finding the optimal labling confidence function and thresholds. In order to find the optimal confidence function, Colander treats the auto-labeling objective as an optimization problem that maximizes the coverage under label error constraint. They use a neural network as the confidence function, and design a surgate for the optimization problem which can then be optimized using gradient-based methods.

**Strengths:**

S1: The paper turns auto-labeling into an optimization problem, and proposes a disciplined solution that can be solved by gradient methods.

S2: The proposed optimization surrogate may be adopted by other auto-labeling methods.

S3: The paper presents extensive experiments to compare with existing methods, and the results are promising.

**Weaknesses:**

W1: the paper is overall well-written, but the discussion of the thresholds is confusing or missing details. Specifically:
- In line 89, it says "the vector _**t**_ denotes scores over _k_ classes", and I suppose score means the predicted scores (or probabilities) of the _k_ class, but later _**t**_ is defined as a vector of thresholds.
- In line 144 (P1), it is unclear what _T^k_ stands for, and how the set of threshold, _T_, is determined.
- In Algorithm 1, the Colander produces the estimated confidence function and thresholds hat _ti'_ (line 14), but the threshold is not used, and it relies on Algorithm 2 to estimate the threshold for each class. There is no discussion what the difference between these two thresholds.

W2: There is no discussion of the computation overhead of Colander.

**Questions:**

Q1: How do you determine the threshold set of _T_? Is it a grid of _k_ classes * _n_ steps in [0,1]?

Q2: Why not use the thresholds produced by Colander (line 14 of Algorithm 1)?

Q3: My understanding is the that confidence function is searched by a neural network using gradient descent, and the thresholds are searched by iterating the training process over the threshold set T. But can we also make the thresholds a neural network (meaning that thresholds would be sample-specific)?

**Limitations:**

The computation cost of Colander increases linearly with the size of the set of confidence thresholds _T_, but it seems the process requires finer granularity in _T_ in order to get a better estimate of the optimal confidence function and thresholds.

---

> ### Author Rebuttal · Authors · 2024-08-06
>
> We appreciate your feedback and acknowledgment of the paper's strengths. Our response:
>
> **Clarification on thresholds**
>
> The vector $\mathbf{t}$ denotes the thresholds over $k$ classes. $T^{k}$ stands for the space of threshold vectors $\mathbf{t}$. Because we solve a relaxed version of the optimization, using  $\hat{\mathbf{t}}'_i$ does not guarantee auto-labeling error below $\epsilon_a$. We estimate the thresholds again in line 16 to ensure the auto-labeling error constraint is strictly followed by the estimated threshold.  We have updated the draft to further clarify this point.
>
>
> **Discussion on computation overhead**
>
> The wall clock time of our method is similar to other post-hoc methods. For instance, a single run on CIFAR-10 setting on NVIDIA RTX A6000 takes roughly 1.5 hours with post-hoc methods and roughly 1 hour without post-hoc methods. We have included a discussion on the computation overhead in the paper.
>
> **Answers**
>
>  **A1.** Yes, our current implementation uses a grid in $[0,1]$ as the set $T$. We emphasize that this is not a necessity. We chose it to favor a simple implementation. It boils down to evaluating the number of thresholds equal to the number of validation points given in the round, so the effective size of $T$ is the number of validation points $N_v$. Moreover, a binary search based implementation can reduce the effective size to $\log(N_v)$ for each class.
>
> **A2.**  See the above discussion on thresholds.
>
>
> **A3.** It is an interesting idea for future works to make thresholds instance-specific. We chose the current thresholding technique as it is backed by the theoretical guarantees on auto-labeling error and coverage [1].
>
>
> **Limitation.** Please see our response A1 above.
>
>
> [1] Vishwakarma et al.,  Promises and Pitfalls of Threshold-based Auto-labeling,  NeurIPS 2023,

---

> ### Comment · Reviewer_ffq7 · 2024-08-12
>
> Thanks for the clarification on the thresholds and the additional experiments! My questions and concerns regarding the proposed method have been addressed and I decided to raise the presentation score.

---

### Official Review · Reviewer_oXkX · 2024-07-14

**Soundness:** 3
**Presentation:** 3
**Contribution:** 3
**Rating:** 7
**Confidence:** 3

**Summary:**

This paper proposes a novel auto-labeling method, called Colander. In contrast to existing works, Colander models the objective of finding an optimal confidence function as a constrained optimization problem (the confidence function should have maximum coverage whilst obtaining a sufficiently low error rate; both components are controlled with a penalty term). Colander is evaluated and compared to baselines on four datasets covering vision and language tasks: MNIST, CIFAR-10, Tiny-ImageNet and 20 Newsgroups. The obtained results clearly indicate that the method improves upon existing baselines, both in terms of coverage and error rate.

**Strengths:**

* The paper proposes a novel approach to identifying confidence functions for auto-labeling. As indicated in the paper, finding confidence functions for auto-labeling is challenging, and this paper tackles this problem elegantly via constrained optimization.
* The paper’s results are promising, showing that the introduced method improves upon existing baselines.
* Overall, I believe that this paper provides a solid contribution to the research area of auto-labeling.

**Weaknesses:**

* The paper focuses heavily on formally introducing Colander, and the experimental content is comparatively thin. To provide the reader with a better notion of robustness, it would for example have been useful to provide additional details on the hyperparameter search in the main paper, and how selecting those differently affects both the coverage and error rate. Likewise, it would have been useful to provide additional details on the impact of the introduced penalty term.
* It would also have been interesting to better understand the relationship between chosen model architecture for a dataset and auto-labeling performance.
* The paper scarcely discusses limitations in the conclusion section, yet I would have expected a more detailed discussion of where and how this approach is limited.
* Related to that, the paper does not touch upon future work / research questions that the obtained results create.

**Questions:**

* Did you conduct additional experiments evaluating how different architectures affect auto-labeling performances for different datasets (i.e., how does changing model architectures for each of the four analyzed datasets affect auto-labeling performance)?

**Limitations:**

As discussed in the Weaknesses, the paper mentions a limitation in the conclusion section but I'd encourage the authors to further elaborate on how and where their approach is limited, and how such limitations can potentially be addressed in future work.

---

> ### Author Rebuttal · Authors · 2024-08-06
>
> Thank you for your insightful review and for recognizing the key strengths of our paper. Our responses:
>
>
> **Details and effects of hyperparameter selection**
>
> Due to space constraints, we deferred a detailed discussion on hyperparameter search to Appendices C.3 to C.7. We plan to incorporate more details in the main paper given the additional space in the camera-ready version. In the rebuttal pdf (Table 5) we provide results with various choices of the penalty term $\lambda$. We observe our method is robust to a wide range for  $\lambda$, except around extreme values.
>
>
> **Effect of model architecture choice**
>
> In TBAL it is not a priori clear what model the practitioner should use. The overall system is **flexible enough to work with any chosen model class**. Our focus is on evaluating the effect of various training time and post-hoc methods designed to improve the confidence functions for any given model. To answer the query, **we ran experiments with Resnet18 and ViT models in the CIFAR-10 setting** (see Table 6 in the rebuttal pdf). As we expected there are variations in the results in the baselines due to model choices but **our method maintains high performance irrespective of the classification model used**. This is due to its ability to learn confidence scores tailored for TBAL.
>
>
> **Limitations and future work**
>
> We discussed these in the conclusions sections and briefly reiterate them here. Colander, like other post-hoc methods, relies on validation data to learn the confidence function. The use of validation data is in general a requirement in TBAL systems as noted in [1]. Reducing or eliminating this dependence is an important direction for future work. It is also related to the general problem of model evaluation in machine learning where solutions based on active testing have emerged [2]. We have updated the draft with additional discussions.
>
> [1] Vishwakarma et al., Promises and Pitfalls of Threshold-based Auto-labeling, NeurIPS, 2023.
>
> [2] Kossen et al., Active Testing: Sample–Efficient Model Evaluation, ICML, 2021.

---

> > ### Comment · Reviewer_oXkX · 2024-08-09
> > **Thanks for the detailed response!**
> >
> > Thank you for providing such a detailed response and the additional results and insights. I will maintain my score and recommend acceptance as indicated in my initial review.

---

### Official Review · Reviewer_Nkxd · 2024-07-22

**Soundness:** 3
**Presentation:** 4
**Contribution:** 2
**Rating:** 6
**Confidence:** 4

**Summary:**

This paper discusses threshold-based auto-labeling functions aimed at identifying a large subset of unlabeled instances where the auto-labeling error remains below a specified threshold. The authors observed that standard temperature-scaled softmax scores are inadequate for effectively thresholding labeled instances. To address this, they propose learning a confidence scoring function and a threshold based on a subset of the validation data. This confidence function is a two-layer neural network trained on the logits from the last two layers of the base model. Extensive experiments were conducted on both image and text datasets using various training-time strategies to optimize the base model, demonstrating the efficacy of their approach.

**Strengths:**

1. The paper is well written, with a variety of experiments conducted to demonstrate the applicability of their proposed method. The choices of strategies and hyperparameters are documented well.
2. The performance of the proposed method is quite significant, achieving lower error rate and higher coverage than the other methods, especially on the harder datasets.

**Weaknesses:**

1. The whole TBAL procedure is a hybrid mixture of iterative training/self training and active learning. While the authors explained the differences between TBAL and self-training + active learning (ST+AL), the difference seems small (mostly, TBAL aims to identify a low-error auto-labelled dataset and ST+AL aims for a good classifier). These two goals do not seem fundamentally different, and can be easily translated to each other. Given the similarity of the framework, I feel it is still necessary to include some ST/AL works in the experiments.
2. A lesser weakness is that due to iterative training, the proposed method and compared methods all went through rounds of data selection, which makes the comparison indirect. For example, it would be hard to tell the exact improvement on thresholding quality the method brings.

**Questions:**

n/a

**Limitations:**

The authors acknowledge that the limitation of their work is the requirement of validation data.

---

> ### Author Rebuttal · Authors · 2024-08-06
>
> Thank you for your detailed feedback and for noting the strengths of our paper. Our responses are the following:
>
>  **On ST/AL works in the experiments**
>
>  Our response involves the following 3 points:
>
>  1. The focus of our paper is to study confidence functions for existing TBAL techniques [1,2,3]. Our main contribution is a method to learn the **optimal confidence function for TBAL**. Given this focus, we have used various natural and widely used choices of confidence functions as baselines. In particular, we have considered both post-hoc methods and train-time methods used for calibrating confidence functions and thoroughly evaluated our proposed method for learning confidence function vs the combination of train-time and post-hoc methods. ST/AL is not a method for learning or calibrating confidence functions, and hence cannot be used as a baseline in our setting.
>
>
>  2. However, given the overlapping components, we addressed the fundamental differences between TBAL and ST/AL in our paper. Our experiments demonstrate that methods like ST/AL, which first aim to learn the best possible classifier from a given function class before performing auto-labeling, **can be severely constrained by the choice of function class**, especially when it does not contain a high accuracy classifier. In contrast, **for TBAL it is not necessary to learn a classifier with high accuracy**. Indeed, TBAL can iteratively auto-label and cover much of a dataset with classifiers that are moderately accurate.
>
>   3. Our aim is **not to establish TBAL as the state-of-the-art method** for data labeling or learning a good model with less data. For this reason, the comparison with ST/AL and other methods is not useful in our work: **what we seek to know is whether TBAL itself can be improved with our new confidence funtion approach**. An extensive evaluation of TBAL, ST/AL, and other similar techniques would be an interesting  work. The differences in the goal -- labeling data with reliability guarantees versus learning the best model in the class with fewer labels are important distinctions and these are discussed in detail along with thorough empirical comparisons in [1].
>
>
> **On indirect comparison due to multiple rounds**
>
> We agree that multiple rounds of auto-labeling and data selection make the comparison indirect. However, it is evident that the gains are due to using Colander since we keep all the other components as it is and only change the confidence functions. To make this more clear, we have run experiments in the single round (passive learning) setting as well, and our results are consistent with the multi-round setting. Reinforcing the fact that the performance improvements are due to Colander, Table 4 in the pdf shows results in single round setting. The evolution of coverage and error over multiple rounds in the multi-round setting is shown in Figure 1.
>
>
> [1] Promises and Pitfalls of Threshold-based Auto-labeling, Vishwakarma et al., NeurIPS, 2023,
>
> [2] MCAL: Minimum Cost Human-Machine Active Labeling, Qiu et al. ICLR, 2023.
>
> [3] AWS Sagemaker Ground Truth https://aws.amazon.com/sagemaker/data-labeling/

---

### Official Review · Reviewer_7Dad · 2024-07-28

**Soundness:** 4
**Presentation:** 4
**Contribution:** 4
**Rating:** 8
**Confidence:** 4

**Summary:**

This paper addresses the challenges of overconfident scores in threshold-based auto-labeling (TBAL) systems. It critiques existing confidence scoring and calibration methods and introduces Colander, a new post-hoc method tailored to optimize TBAL performance.

**Strengths:**

-	The paper has a good identification of the problem which is over confidence of existing TBAL confidence functions, and it proposes a novel framework to find the optimal confidence function.
-	The paper conducts extensive experiments on both image and text data.
-	The paper compares various post-hoc functions.
-	The paper is well written with great details in Appendix

**Weaknesses:**

-	The authors rightly mention limitations (“A limitation of Colander is that similar to other post-hoc methods it also requires validation data to learn the confidence function. Reducing (eliminating) this dependence on validation data could be an interesting future work.”) in the conclusion. So, does this mean that they use Gold data for validation? If that is the case, that might be the weakness of TBAL in general. The following paper talks about it in detail: https://aclanthology.org/2023.acl-long.796/

**Questions:**

-	Not a question but authors can consider following related paper for comparison. It also works with confidence scoring: https://aclanthology.org/2021.naacl-main.84.pdf

**Limitations:**

Yes.

---

> ### Author Rebuttal · Authors · 2024-08-06
>
> Thank you for your positive and encouraging review. We are delighted with your assessment and recognition of our work’s contributions. Our response:
>
> **Dependence on validation data.**  Yes,  Colander, like other post-hoc methods, relies on validation data to learn the confidence function. In fact, this is in general a requirement in TBAL systems as noted in [4] and your reference [1]. Reducing or eliminating this dependence is a useful direction for future work. It is also related to the general problem of model evaluation in machine learning where solutions based on active testing have emerged [3].
>
> **Question on a related work.**  We appreciate the suggestion of the related paper [2]. This work uses softmax scores in a similar pipeline augmented with external knowledge sources. In our experiments, we have a comparison with the softmax scores. On the other hand, we believe using scores from Colander could be helpful in [2]. We have included a discussion on this idea in the draft.
>
> [1] https://aclanthology.org/2023.acl-long.796.pdf
>
> [2] https://aclanthology.org/2021.naacl-main.84.pdf
>
> [3] http://proceedings.mlr.press/v139/kossen21a/kossen21a.pdf
>
> [4] https://openreview.net/pdf?id=RUCFAKNDb2

---

> > ### Comment · Reviewer_7Dad · 2024-08-07
> > **Thank You!**
> >
> > Thank you for your response. I enjoyed reading your paper and learned a lot.
> >
> > A few things regarding the reviewer YvdP. As I differ a lot in scores, I felt I should comment on a few of them.
> >  - The presentation of the paper is really good. I do believe that Figure 1 was useful while I was reading the paper and two of my colleagues also agree with that.
> >  - Many comments are unreasonable for example "There is no theoretical analysis and understanding of mathematical properties"
> >
> > Good luck with the other reviewers!

---

### Official Review · Reviewer_YvdP · 2024-07-31

**Soundness:** 3
**Presentation:** 1
**Contribution:** 2
**Rating:** 4
**Confidence:** 3

**Summary:**

This paper introduces a Colander for threshold-based auto-labeling (TBAL) pipeline. Colander trains a flexible confidence function using latent information of the classifier model, instead of fixed confidence functions. The optimization problem of Colander is based on the objective of TBAL, that is, maximizing coverage while minimizing auto-labeling errors. The authors consider differentiable surrogates for the 0-1 variables using the sigmoid function to make this problem tractable using gradient-based methods.

**Strengths:**

This paper is well-motivated. The proposed method is explained with simplicity and clarity.

**Weaknesses:**

Contribution: Although this paper proposes the Colander for the TBAL pipeline, there is no theoretical analysis and understanding of mathematical properties.

Completeness:
- The paper includes unknown notation, which is used without pre-defined.
- In Section 3.3, an undefined procedure exists: RandomQuery, RandomSplit, and ActiveQuery. In the case of Random method, it is possible to guess by readers, but in the case of ActiveQuery, a detailed explanation is needed.
- There are a few expressions that make it difficult to figure out the intent. Additionally, some keywords used in the paper need to be unified.
   - What is the difference between the score (or scoring) function and the confidence function?
   - What is the difference between ‘inherent tension’ (in line 33) and trade-off?
   - What is the difference between post-hoc method and calibration method?
- The reviewer is unsure if Figure 1 is necessary for the reader: it contains acronyms that are not defined in advance, such as ERM, and certain formulas (h) are left undefined in the introduction.
- Line 174, line 186: The reader does not know which Appendix to look at because Appendix information is omitted.
- Appendix, Algorithm 2, line 8: The formula where C_1  weight is considered is not defined in the text.
- Appendix, Table 3: there are typos.
- This paper is considered to have no significant contributions compared to previous research about TBAL[1].
[1] Vishwakarma, Harit, et al. "Promises and pitfalls of threshold-based auto-labeling," Neural Information Processing Systems, 2023.

**Questions:**

Active Learning Vs. TBAL
- Active learning and TBAL fundamentally play the role of labelers for exploiting unlabeled raw data. Lines 100-104 include these differences, but it is hard to get the point. Please explain these differences in detail.
- Although the paper includes a performance comparison with active learning (and self-training), an algorithmic setup of active learning is not provided. Please provide experimental settings for understanding whether it is fair or not.
- Why does the TBAL process require a human labeler at every iteration? Is it impossible to use labeled and unlabeled data completely separately in advance?

Colander (proposed solution)
- In Algorithm 1 (line 14), Colander extracts a confidence function  g ̂_i and t ̂_i^' through an optimization problem. As the reviewer’s knowledge, t ̂_i^' is different from  t ̂_i obtained estimation threshold process. Where is t ̂_i^' used?
- The optimization problem (P1,2,3) is solved using a given classifier. It means that this optimization process only works when the classifier's performance is guaranteed. Is it right and is this a correct assumption?
- The reviewer understood that the proposed solution configures the confidence function as a neural network, so it is flexible. But what is the meaning of 'choice of confidence function.' Does Colander learn multiple confidence functions and choose one of them?
- Why did the authors select exactly two layers of the classifier as inputs to the confidence function? There are various options: single layer, three layers, and inclusion of output information.
- Is it right to learn a new classifier/confidence function every round? If so, how long does the entire process require? If not, some expressions in the algorithm (pseudocode) and Section 3.3 need to be changed.

Post-hoc method
- Most baseline methods were proposed before 2020, except one algorithm. Please check following methods.
  -	R. Verma, et al. "Learning to defer to multiple experts: Consistent surrogate losses, confidence calibration, and conformal ensembles." AISTATS, 2023.
  -	L. Tao, et al. "Calibrating a deep neural network with its predecessors." IJCAI, 2023.
  -	T. Joy, et al. "Sample-dependent adaptive temperature scaling for improved calibration." AAAI, 2023.
  -	The reviewer would appreciate it if the authors could provide some insight/intuition into the problems with previous methods. The experimental sections just show empirical results and lack discussion.

Empirical results
- It is important to provide accurate settings when performing partial improvements in the overall pipeline. Have you checked the performance of various procedure method changes such as ActiveQuery?
- Are there any performance analysis results regarding changes in confidence scores?
- The experimental results clearly show that the algorithm has high performance in the provided experimental settings. What about time efficiency? Have you checked the wall-time?
- In the optimization problem (P3), how did the authors select the upper-bound error parameter ϵ? How do experimental results change depending on parameter changes?
- The value of C_1 in line 8 of Algorithm 2 seems to be fixed at 0.25. What results can see when it changes?

**Limitations:**

In this paper, there was not enough discussion about the limitations.

---

> ### Author Rebuttal · Authors · 2024-08-07
>
> We appreciate the detailed review. We have updated our work to account for points on notation and writing. Our response:
> ### Contributions
> We believe the reviewer has missed our paper's key contribution, specifically **with respect to prior work on TBAL [1]**. Our work **does not reiterate [1]**. Instead, *the weaknesses of the TBAL technique in [1] inspired our work*. That is, [1] uses a simple, fixed confidence function that leads to poor performance in terms of error rate and coverage. Our goal is to understand the impact of this choice and to produce **innovations that resolve these weaknesses**, as illustrated in our draft's Figure 2. This is important given wide industry adoption of TBAL [2]. We:
> - demonstrate the importance of confidence functions in TBAL, evaluate several choices of functions, and find them to be **severely limited---as they are not well-aligned with TBAL objectives**.
> - propose a framework to learn the optimal confidence function and provide a practical version of it.
> - provide extensive empirical evaluations, showing that using confidence functions learned via Colander in TBAL improves auto-labeling coverage significantly.
> ### Theoretical Analysis
> Our main contribution lies in demonstrating the issues with common choices of confidence functions in TBAL, proposing a principled solution to learn the optimal confidence functions for TBAL, and showing its effectiveness empirically. A rigorous theoretical analysis of our method is left as future work.
> ### Completeness
> * We have fixed notation and typos.
> * ActiveQuery is discussed in Appendix B.3 (lines 583-603). We have added a detailed algorithm.
> * We use scoring and confidence functions synonymously.
> * The terms inherent tension and trade-off indeed refer to similar behaviors.
> * Train-time and post-hoc calibration methods are standard terms: Train-time methods modify the training procedure while post-hoc methods operate on trained classifiers.
> * Fig. 1 shows a simplified workflow of the standard procedure highlighting the role of confidence functions. We have set out to make the figure readable without the notations as well; we kept the notations for consistency with later figures.
> * It is the Appendix B.2 (line 571 onwards)
> * $\hat{\sigma}$  is the standard deviation of the error estimate. We have included the formula.
> ### Active Learning vs. TBAL
> Our answer:
> 1. The details of the AL setup and experiment are in Appendix A.1. We added further detail.
> 2. The aim of the AL experiment is to **highlight the fundamental difference between TBAL and procedures (such as AL)** that seek to first learn the best possible classifier from the given function class and then do auto-labeling. These methods are severely limited by the choice of function class. On the other hand, TBAL succeeds since it iteratively auto-labels and eliminates the auto-labeled space. The comparison is fair as the methods use same amount of labeled data, same function classes, and training to learn the model.
> 3. We adhered to the workflow of TBAL from [1]. While other ways to involve the human labeler are possible, these would not permit a comparison to [1].
> ### On Colander
> 1. Because we solve a relaxed version of the optimization, $\hat{\mathbf{t}}'_i$ is not guaranteed to ensure auto-labeling error below $\epsilon_a$. We estimate the thresholds again in line 16, to ensure the auto-labeling error constraint is strictly followed by the estimated threshold.
> 2. The optimization process does not require any specific performance assumptions on the classifier. It has to be better than a random classifier to get any meaningful output from the procedure; it is not a strong requirement.
> 3. Colander does not learn multiple functions to choose from. In each round, it takes the current classifier and learns a confidence function by solving P3.
> 4. Colander can use any function class for $g$. In experiments, we chose 2-layer nets and successfully used the same across all datasets, thus we may not need an exhaustive architecture search. Intuitively, we do not need a large network for $g$ since $h$ already performs the heavier representation learning work. As a result, simple models are preferable to avoid overfitting and to reduce training time (since post-hoc methods should be fast).
> 5. TBAL learns a new classifier in every round, thus the confidence function also needs to be relearned. We comment on the time below.
> ### Baselines
>  We use prominent and recent post-hoc and train-time baselines. Thank you for the references. Of these [3] fits our setting. We provide results in the attached pdf (Table 1); these are consistent with the draft. The key insight is that baselines are not tailored to the TBAL objective. Colander is designed to maximize TBAL performance. Figure 2 and its discussion make this point.
>
> ###  Empirical results
> 1. Our primary focus is on confidence functions, and to avoid confounding, we have chosen not to compare various active querying strategies within this work. Including multiple variables would obscure the effects of our method.
> 2. Could you please clarify this question?
> 3. The wall clock time of our method is similar to other post-hoc methods: a single CIFAR-10 run on an NVIDIA RTX A6000 is roughly 1.5 hours (post-hoc) and 1 hour (no post-hoc).
> 4. Our framework is flexible to work with any choice of $\epsilon_a \in (0,1)$. In experiments, we used a fixed $\epsilon_a=0.05\$. We provide additional results in Table 2 in the pdf with varying $\epsilon_a$ to demonstrate its effect.
> 5. Results with $C_1 \in ${$0.0, 0.25$} in 20-newsgroup setting with vanilla train-time and all post-hoc methods are in Table 3. $C_1=0.0$ leads to higher variance in auto-labeling error, consistent with previous works [1].
>
> [1] Vishwakarma et al., Promises and Pitfalls of Threshold-based Auto-labeling, NeurIPS 2023.
>
> [2] https://aws.amazon.com/sagemaker/data-labeling
>
> [3] Joy et al., Sample-dependent Adaptive Temperature Scaling for Improved Calibration, AAAI 2023.

---

> > ### Comment · Reviewer_YvdP · 2024-08-11
> >
> > Thank the authors for their hard work! After an additional explanation, some of my concerns were addressed. So, I have raised my score.
> >
> > Clarification regarding my score.
> > +) This work is well-motivated. I believe the community would appreciate this work, which can be very practical. Additional results provided in the rebuttal period strengthen the claim.
> > -) The manuscript is not ready to be published. Notations can be revised, and the figure can be better described. The text size in the figure is too tiny.  The critical limitation of this work is outdated baselines. The authors added a new baseline in the rebuttal period, but overall evaluations are quite limited.
> >
> > So, I think this work is around the borderline, but I lean to the reject side. In terms of presentation and evaluation, I strongly believe this manuscript is not ready to be published.

---

### Author Rebuttal · Authors · 2024-08-07

We thank all of the reviewers for their insightful and positive feedback. We have used their suggestions to improve our draft, added experiments, and improved the clarity of our work. Before providing individual responses, we (1) summarize the strengths highlighted by reviewers, (2) provide a pair of common responses, and (3) describe some new experiments that strengthen our work.


## 1. Strengths

**Motivation and novelty of our method (7Dad, Nkxd, oXkX, ffq7, YvdP):** Reviewers appreciated our **novel and well-motivated approach** to addressing the issue of overconfidence in auto-labeling tasks. The reviewers appreciated our mathematically grounded framework to learn the optimal confidence function for TBAL. They also noted our optimization framework could be useful for a variety of auto-labeling methods.

**Significant performance improvements over the vanilla TBAL (Nkxd, oXkX, ffq7):** Integrating our method into TBAL provides significant performance improvements over the vanilla version used in the previous works on TBAL [1].

**Thorough empirical evaluation (7Dad, Nkxd, ffq7):** The evaluations covered diverse datasets, including text and image data, showcasing the proposed method's applicability across different domains. We provide comparisons against several train-time and post-hoc methods commonly used to reduce the overconfidence issue.

**Clarity (7Dad, Nkxd, oXkX):** Most reviewers found our paper well-written and easy to follow, and they appreciated the illustrations. They also liked our step-by-step discussion for designing the objective function, which leads to a tractable optimization problem.

## 2. Response to common questions
* **On contributions and practical significance**. We are motivated by the fact that TBAL is widely used in practice to create labeled datasets [2], including by industry giants like Amazon (in its SageMaker Ground Truth product). TBAL has also recently been the subject of theoretical study in exciting works such as [1,3]. Given this interest---both practical and theoretical---our goal is to understand the role of **confidence functions, a key component in TBAL that has not been previously studied**. As our work shows, TBAL systems are heavily affected by the choice of confidence function. This led us to develop a new approach for these that **leads to substantial improvements in TBAL systems**. Given its strengths, we anticipate that our new method will become the standard for TBAL systems.

* **On active learning/self-training and TBAL**: The focus of our work is on understanding and improving confidence functions for TBAL. A prior work [1] studies TBAL theoretically and shows the fundamental differences between active learning and TBAL. It also provides extensive experiments illustrating this difference. Since there are some overlaps in these methods, we have a brief discussion and a simulation in our paper to clarify the fundamental differences between them.


## 3. Additional Experiments
We provide a brief summary of the additional experiments and results here. The tables and figures are in the attached rebuttal pdf. The details are deferred to individual reviewer responses.

1. (YvdP) We run TBAL with the additional calibration baseline Adaptive Temperature Scaling (AdaTS) [4] and report the results in Table 1. The results are **consistent with the main paper and our expectations**.
2. (YvdP) We run TBAL with five values of $\epsilon_a \in$ { $0.01,0.025,0.05,0.075, 0.1 $ } and report the results in Table 2. **As expected** the auto-labeling error is high with larger values of $\epsilon_a$ and smaller with small $\epsilon_a$.

3. (YvdP) The results with $C_1 \in$ { $0.0, 0.25$ } are in Table 3. Using $C_1=0.0$ leads to higher variance in the auto-labeling error. This is consistent with prior work [1].

4. (Nkxd) We further demonstrate that the **performance gains are due to the use of Colander**, even if methods use multiple rounds. To do so, we show the evolution of coverage and error over multiple rounds in Figure 1 in the rebuttal pdf. The effects of using Colander are visible from the first round itself, and the following rounds improve performance further. We also run a single round (passive) variant of TBAL where we sample all the human-labeled points for training ($N_t$) randomly at once, train a classifier, do auto-labeling, and then stop. This setting avoids confounding due to multiple rounds. We observe that using Colander yields significantly higher coverage in comparison to the baselines (see Table 4 in the pdf). This reinforces the fact that the gains in the multi-round TBAL are directly due to Colander, while multiple rounds of data selection, training, and auto-labeling are superior to doing everything in a single round.


5. (oXkX) We run TBAL with ViT and ResNet18 models. The results are in Table 6. We see the choice of classification model affects baseline performance but TBAL with Colander remains robust to the model choice.

6. (oXkX) The results with $\lambda$ variation are in Table 5. We see Colander is robust to a wide range of values of $\lambda$, except extreme values.


**References**

[1] Vishwakarma et al., Promises and Pitfalls of Threshold-based Auto-labeling, NeurIPS 2023.

[2] https://aws.amazon.com/sagemaker/data-labeling/

[3] Qiu et al., MCAL: Minimum Cost Human-Machine Active Labeling, ICLR, 2023

[4] Joy et al., Sample-dependent Adaptive Temperature Scaling for Improved Calibration, AAAI 2023.

---

### Comment · Area_Chair_nZcx · 2024-08-13
**Please be polite during the discussion**

Dear reviewers,

Thanks for responding to the authors' rebuttals! Please try and be polite during discussions. Everyone has a right to have their own opinions on the submission, and nobody should be pressured to change the review or rating.

Thanks!

---

### Decision · Program_Chairs · 2024-09-25

**Decision:**

Accept (poster)

**Comment:**

The submission proposed a principled approach to optimize for confidence thresholds for TBAL, and showed its effectiveness empirically. Reviewers pointed out that there is no rigorous theoretical analysis or results, but generally agreed that the approach is principled and empirical results are sound.

There were considerable discussions on the difference between TBAL and active learning, and I think the authors have done sufficient work in trying to clarify the differences between the two approaches, and added sufficient details to the submission. TBAL is a relatively new and interesting framework with some theoretical properties analyzed by previous work, e.g. [1]. I think this paper proposed an interesting approach to improve empirical TBAL performance.

[1] Vishwakarma et al., Promises and Pitfalls of Threshold-based Auto-labeling, NeurIPS 2023.